# Asymptotics of SGD in Sequence-Single Index Models and Single-Layer Attention Networks

**Luca Arnaboldi**
IdePhics Laboratory
EPFL
Lausanne, Switzerland

**Bruno Loureiro**
Département d'Informatique
École Normale Supérieure - PSL
Paris, France

**Ludovic Stephan**
ENS AI
University Rennes
Rennes, France

**Florent Krzakala**
IdePhics Laboratory
EPFL
Lausanne, Switzerland

**Lenka Zdeborová**
SPOC Laboratory
EPFL
Lausanne, Switzerland

## Abstract

We study the dynamics of stochastic gradient descent (SGD) for a class of sequence models termed Sequence Single-Index (SSI) models, where the target depends on a single direction in input space applied to a sequence of tokens. This setting generalizes classical single-index models to the sequential domain, encompassing simplified one-layer attention architectures. We derive a closed-form expression for the population loss in terms of a pair of sufficient statistics capturing semantic and positional alignment, and characterize the induced high-dimensional SGD dynamics for these coordinates. Our analysis reveals two distinct training phases: escape from uninformative initialization and alignment with the target subspace, and demonstrates how the sequence length and positional encoding influence convergence speed and learning trajectories. These results provide a rigorous and interpretable foundation for understanding how sequential structure in data can be beneficial for learning with attention-based models.

Stochastic Gradient Descent (SGD) is the core optimization tool driving modern machine learning. Recent years have seen substantial progress in understanding its dynamics, particularly in two-layer networks [Saad and Solla, 1995, Mei et al., 2018, Chizat and Bach, 2018, Rotskoff and Vanden-Eijnden, 2022, Sirignano and Spiliopoulos, 2020, Arnaboldi et al., 2023a]. While global convergence is qualitatively well-understood when the network is wide enough, quantitative results are scarcer. A particularly fruitful body of recent theoretical work addressing this gap has focused on deriving precise convergence rates for particular model classes on synthetic data, such as high-dimensional Gaussian single and multi-index models [Ben Arous et al., 2021, Abbe et al., 2022, 2023]. These advances have sparked a wave of follow-up studies [Damian et al., 2022, 2023, Dandi et al., 2024, Bietti et al., 2023, Ba et al., 2023, Moniri et al., 2023, Mousavi-Hosseini et al., 2023, Zweig and Bruna, 2024, Berthier et al., 2024, Arnaboldi et al., 2024a,b], deepening our understanding of what problems are hard to learn for neural networks trained under SGD.

While multi-index models have served as a cornerstone in theoretical analyses of learning, they remain far from the architectures driving recent breakthroughs in machine learning. Modern advances in learning from sequential data — particularly in natural language processing — are increasingly dominated by attention-based models such as the Transformer architecture [Vaswani et al., 2017]. These models introduce a paradigm shift through self-attention mechanisms, which dynamically reweight the influence of each input token based on its relevance to others. Through successive layers of attention, Transformers capture intricate dependencies across sequences, enabling state-of-the-art

39th Conference on Neural Information Processing Systems (NeurIPS 2025).

performance in tasks ranging from machine translation to large-scale language modeling [Brown et al., 2020, Kenton and Toutanova, 2019].

The main goal of this work is to extend our theoretical understanding of learning with SGD on multi-index models to attention-based architectures and sequential data. Inspired by Cui et al. [2024], Troiani et al. [2025], our focus will be on the following class of single-layer, tied attention model:

$$\boldsymbol{f}_{\boldsymbol{w}}(X) = R\left[\mathrm{softmax}\left(\left(X + \frac{P}{\sqrt{d}}\right)\boldsymbol{w}\boldsymbol{w}^{\top}\left(X + \frac{P}{\sqrt{d}}\right)^{\top}\right)\right]. \tag{1}$$

where $\boldsymbol{w} \in \mathbb{R}^d$ are the trainable weights and $R$ is the *reduction map*, which allows passing from a $L \times L$ matrix to a $k$-dimensional vector. As detailed in Appendix A, this model is a reduction of the standard attention mechanism, where (i) key and query matrix are tied, and $d_{\mathrm{head}} = 1$: $Q = K = X \cdot \boldsymbol{w} \in \mathbb{R}^{L \times 1}$ ; (ii) since we are considering a single layer attention, and we do not need to learn a new representation of the sequence, the value matrix is the identity: $V = I_L$. In this work, we will be interested in the optimization properties of the model in eq. (1) when trained under (spherical) one-pass stochastic gradient descent (SGD) from a random initial condition $\boldsymbol{w}^0 \sim \mathrm{Unif}(\mathbb{S}^{d-1}(\sqrt{d}))$:

$$\boldsymbol{w}^{\tau+1} = \frac{\boldsymbol{w}^{\tau} - \gamma\nabla_{\boldsymbol{w}}\ell(X^{\tau}, \boldsymbol{y}^{\tau}; f_{\boldsymbol{w}})}{\left\|\boldsymbol{w}^{\tau} - \gamma\nabla_{\boldsymbol{w}}\ell(X^{\tau}, \boldsymbol{y}^{\tau}; f_{\boldsymbol{w}})\right\|}\|\boldsymbol{w}^{\tau}\|. \tag{2}$$

with the squared loss:

$$\ell(X, \boldsymbol{y}; f_{\boldsymbol{w}}) = \left\|\boldsymbol{y} - f_{\boldsymbol{w}}(X)\right\|_{\mathrm{F}}^2 = \sum_{i=1}^{k}\left(y_i - f_{\boldsymbol{w}}(X)_i\right)^2. \tag{3}$$

Note that for one-pass (a.k.a. *online* or *streaming*) SGD each sample is only seen once, meaning that the sample complexity of the algorithm coincides with the convergence rate. The spherical constraint is considered to simplify the mathematical analysis, a common assumption in the analysis of SGD for single-index models [Ben Arous et al., 2021, Damian et al., 2022].

To derive a sharp characterization of the sample complexity and convergence rate of SGD for the single-layer attention mechanism in eq. (1), we assume that the sequence data $(X, \boldsymbol{y})$ is generated from the following Gaussian *sequence single-index* (SSI) model:

**Assumption 1** (Data distribution). *We assume training data* $(X, \boldsymbol{y}) \in \mathbb{R}^{L \times d} \times \mathbb{R}^k$ *is independently drawn from a Gaussian Sequence Single Index (SSI) model:*

$$\boldsymbol{f}_{\boldsymbol{w}_\star}^{\mathrm{SSI}}(X) = \boldsymbol{g}(X \cdot \boldsymbol{w}_\star) \tag{4}$$

*where* $X \in \mathbb{R}^{L \times d}$ *is a Gaussian matrix with entries* $\mathcal{N}(0, 1/d)$, $\boldsymbol{g}\colon \mathbb{R}^L \to \mathbb{R}^k$ *is a vector-valued link function that depends only on the scalar product of the input with a fixed vector* $\boldsymbol{w}_\star \in \mathbb{S}^{d-1}(\sqrt{d})$ *and* $k$ *is a integer that does not depend on* $d$ *nor* $L$.

Recent work by [Cui et al., 2024, Troiani et al., 2025, Cui, 2025] has shown that the single-layer tied attention model in eq. (1) is a particular instance of a class of *sequence multi-index (SMI) models*, creating a bridge between single-index analysis and attention-based learning. This mapping implies that model eq. (1) can learn at best a predictor in this class, justifying the choice for training data. The SSI model class was introduced by Cui et al. [2024] in the context of studying phase transitions for the attention mechanism. These results position the SMI model as a signature synthetic model to study the interplay between attention mechanisms, data structure, and the dynamics of learning algorithms. Despite this progress, the learning dynamics of SMI models under SGD remain unexplored. Prior studies [Cui, 2025, Cui et al., 2024] analyzed the empirical risk minimizer through the heuristic replica method, while [Troiani et al., 2025] rigorously studied the Bayes-optimal estimator for SMI models. However, a theoretical understanding of the population landscape and SGD dynamics in this model is missing. Our work addresses this gap, providing the first rigorous characterization of SGD dynamics in SMI models by leveraging techniques developed for single and multi-index models.

**Main results —** Our main methodological contribution is the generalization of analytical tools to study multi-index models to variants that process sequences of tokens rather than simple vector inputs. Our contributions can be summarized as follows:

- We introduce the notion of *sequence information exponent* (SIE), as a generalization of the information exponent for single-index models Ben Arous et al. [2021]; the SIE has a direct correspondence with the sample complexity of SGD. We also discuss the implications of the positional encoding on the sample complexity, proving it could help SGD to learn faster.

- We analyze the speed-up introduced by the attention mechanism when learning sequential data, compared to models not adapted to treat sequence structures, e.g. fully connected networks. For many problems, we show that the gain is proportional to the sequence length $L$ and in some cases even larger.

- We investigate the interplay between the positional and semantic structure of the data following the setting from Cui et al. [2024], showing that SGD dynamics is not always able to disentangle the two and that a rich phase diagram arises describing the structure of the corresponding population loss and the performance of SGD.

All our formal claim are supported by rigorous proofs, as well as numerical experiments; the code developed is available at `https://github.com/IdePHICS/Sequence-Single-Index`.

**Further Related works —**  There have much activity discussing **SGD with synthetic data on multi-index models** over the last decades or so, see e.g. [Ben Arous et al., 2021, Veiga et al., 2022, Arnaboldi et al., 2023b, Collins-Woodfin et al., 2024, Marion and Berthier, 2023, Montanari and Urbani, 2025] and reference there in. **Information** and **generative** exponent have been the topic of many works over the last few years [Ben Arous et al., 2021, Abbe et al., 2022, Damian et al., 2024, Troiani et al., 2024]. Here we discuss and adapt these notions for sequence models Troiani et al. [2025].

On the topic of the **Theory of SGD in transformers**, Wu et al. [2023] convergence guarantees on SGD for the single layer transformer. Song et al. [2024] also study GD convergence in simple architectures and highlight the existence of suboptimal local solutions. Li et al. [2025] point out that rapid convergence does not guarantee meaningful learning. Li et al. [2023] give sample complexity bounds for a shallow vision transformer. Zhang et al. [2025] being overfitting in SGD trained transformer. Yüksel and Flammarion [2025] focus on gradient-based dynamics for next token prediction tasks. Compared to these work we move beyond convergence results to study how the data structure, e.g. sequence structure and positional encodings, affect sample complexity and behaviours of the the population loss and recovery dynamics in the high-dimensional limit.

Authors of Marion et al. [2024] introduce a model that can be seen as a sequence two-index model. While we assume the input data to be iid Gaussian they assume a spiked covariance structure in the data which makes their results not directly comparable to ours. We anticipate that our results can be generalized to their setting. Another recent work Mousavi-Hosseini et al. [2025] looked at sample complexity separation between attention-based networks and more traditional architectures, while their setting is different, this question is related to ours.

## 1   Setting and definitions

Let $(X^\tau, \boldsymbol{y}^\tau) \in \mathbb{R}^{L \times d} \times \mathbb{R}^k$ denote $i = 1, \ldots, n$ samples drawn from the Gaussian sequence single-index model with weights $\boldsymbol{w}_\star \in \mathbb{R}^d$ and link function $\boldsymbol{g}$, defined in eq. (4). As motivated in the introduction, our goal in this work is to characterize the sample complexity of learning the sequence task $(X^\tau, \boldsymbol{y}^\tau)$ with a tied single-layer attention trained under one-pass (spherical) SGD defined in eq. (2). Note that while the model might appear as too simplified because of the lack of correlation between the tokens, we will show that it is sufficient to capture the main features of sequence models.

As shown in the classical result by Robbins and Monro [1951], one-pass SGD can be understood as a noisy discretization of gradient flow on the *population risk* (often refereed also as *population loss*):

$$R(\boldsymbol{w}) = \mathbb{E}_{X \sim \mathcal{N}(0, I_d/d)} \left[ \ell(X, \boldsymbol{f}_{\boldsymbol{w}_\star}^{\mathrm{SSI}}(X); \boldsymbol{f}_{\boldsymbol{w}}) \right]. \tag{5}$$

Therefore, in order to understand the dynamics in eq. (2) it is important to understand the landscape of the risk above. The key property of single-index models underlying the convergence rate analysis of Ben Arous et al. [2021] is that rotation invariance of the population risk implies that it only depends on a single parameter: the correlation between the target weights and the predictor weight, also

known as an *order parameter* or *sufficient statistic*. A similar property holds for the family of SSI models defined by Assumption 1. Indeed, conditionally on the weights, the following projections

$$\boldsymbol{z}_\star = X \cdot \boldsymbol{w}_\star \in \mathbb{R}^L \quad \text{and} \quad \boldsymbol{z} = \left( X + \frac{P}{\sqrt{d}} \right) \boldsymbol{w} \in \mathbb{R}^L, \tag{6}$$

define joint Gaussian variables, which can be fully characterized by their means and covariances:

$$\mathbb{E}\left[ z_{\star,i} \right] = 0, \quad \mathbb{E}\left[ z_i \right] = e_i$$
$$\text{Cov}\left( z_{\star,i}, z_{\star,j} \right) = \delta_{ij}, \quad \text{Cov}\left( z_i, z_j \right) = \delta_{ij}, \quad \text{Cov}\left( z_{\star,i}, z_j \right) = \delta_{ij} m, \tag{7}$$

where we have introduced the *sufficient statistics*:

$$m = \frac{\boldsymbol{w}_\star^\top \boldsymbol{w}}{d} \quad \text{and} \quad \boldsymbol{e} = \frac{P\boldsymbol{w}}{\sqrt{d}}. \tag{8}$$

These play exactly the same role as the overlap between target and predictor weights in the standard single-index model. Therefore, the population risk can be written as a function of these statistics:

$$R(\boldsymbol{w}) \equiv R(\boldsymbol{e}, m) = \mathbb{E}_{(\boldsymbol{z}_\star, \boldsymbol{z})} \left[ \left\| \boldsymbol{g}(\boldsymbol{z}_\star) - \boldsymbol{R}\left[ \text{softmax}\left( \boldsymbol{z}\boldsymbol{z}^\top \right) \right] \right\|_F^2 \right]. \tag{9}$$

Note that this formulation reduces the problem of understanding the landscape geometry of $R$ in the $\boldsymbol{w} \in \mathbb{R}^d$ space to understanding it in $(\boldsymbol{e}, m) \in \mathbb{R}^{L+1}$ — a significant simplification when the token size $d$ is large with respect to the sequence length $L$, the regime we will focus in this work. Note that, given the fixed norm constraint on $\boldsymbol{w}$, the sufficient statistics are constrained inside the unit ball of $\mathbb{R}^{L+1}$: $\left\| (\boldsymbol{e}, m) \right\| \leq 1$.

**Escaping mediocrity —**  As previously discussed, studying the convergence rate of one-pass SGD is akin to studying the population risk landscape. In the standard single-index model, the picture arising from [Ben Arous et al., 2021, Arnaboldi et al., 2023c] is rather simple: the only critical points of the population risk are a single global minima at the target weights and (possibly) a strict saddle at zero correlation. Therefore, the convergence rate of one-pass SGD from random initialization is dominated by the time taken to escape this saddle-point, a scenario a scenario commonly referred to as *escaping mediocrity* [Arnaboldi et al., 2023c].

As we shall see, the risk landscape of sequence models is richer, with in particular the presence of local minima. Nevertheless, these models share the common property of mediocrity at initialization, with the convergence rate dominated by the flatness of the initial saddle-point. Therefore, we start our discussion by formalizing this notion in the context of SSI models. In the high-dimensional scenario where $d$ is large, the initial weight $\boldsymbol{w}^0$ is approximately orthogonal to the target direction $\boldsymbol{w}_\star$, as well as the positional embedding $P$. Quantitatively, the sufficient statistics are distributed as

$$\lim_{d \to +\infty} \sqrt{d}(\boldsymbol{e}^0, m^0) \sim \mathcal{N}(0, I_{L+1}). \tag{10}$$

Namely, the initial value of the sufficient statistic is $(\boldsymbol{e}^0, m^0) \approx (\boldsymbol{0}, 0)$, with fluctuations of order $O(1/\sqrt{d})$. For the model in eq. (1), $(\boldsymbol{e}, m) = (\boldsymbol{0}, 0)$ is a saddle-point of risk, and the dynamics is divided in two phases:

- The escape from the initial condition, where the model develops a weak correlation with the target direction $\boldsymbol{w}_\star$ and/or the positional embedding $P$;
- Full recovery where it reaches a complete overlap with either the target direction $\boldsymbol{w}_\star$ and/or the positional embedding $P$: $\left\| (\boldsymbol{e}, m) \right\| \approx 1$.

As previously discussed, the first phase is the one that requires the most number of gradient steps Ben Arous et al. [2021], Arnaboldi et al. [2023c]: the sample complexity required for the first phase is always greater or equal to the one required for the second phase; after having reached a small correlation with the target, the attention decay exponentially fast to a complete overlapped state.

**Definition 1** (Weak recovery)**.** *Let $\eta \in (0, 1)$ a parameter independent from $d$. We say that the model has* weakly recovered *the target when $\left\| (\boldsymbol{e}, m) \right\| \geq \eta$. The weak recovery time is then*

$$\tau_\eta^{\text{weak}} = \min \left\{ \tau \geq 0 \colon \left\| (\boldsymbol{e}^\tau, m^\tau) \right\| \geq \eta \right\}.$$

We use this definition of weak recovery as a proxy for identifying the learning has happened, since the subsequent *strong recovery* will be faster. Figure 1 shows some examples of population loss surface: apart from initialization, there are no other critical points where the dynamic can get slowed down.

Finally, for simplicity of the discussion we will make the following assumption on the spherical one-pass SGD dynamics in eq. (2).

**Assumption 2** (Gradient flow approximation). *We approximate the training dynamics of eq.(2) via the following ODE, which corresponds to an order-2 Taylor expansion in $\gamma$:*

$$\frac{d\boldsymbol{w}}{dt} = \mathbb{E}\Big[\nabla_{\boldsymbol{w}}^{\perp}\ell(X, y, f_{\boldsymbol{w}})\Big] - \frac{\gamma}{2}\,\mathbb{E}\Big[\Big\|\nabla_{\boldsymbol{w}}^{\perp}\ell(X, y, f_{\boldsymbol{w}})\Big\|^2\Big]\boldsymbol{w}, \tag{11}$$

*where $\nabla_{\boldsymbol{w}}^{\perp} = (I - \boldsymbol{w}\boldsymbol{w}^{\top})\nabla_{\boldsymbol{w}}$ is the spherical gradient. The time scaling corresponds to $t = \tau/\gamma$.*

As shown in Ben Arous et al. [2021], Arnaboldi et al. [2024c], such an ODE captures both the right weak recovery time for a fixed $\gamma$, as well as the maximal value of $\gamma$ for which the dynamics do not stay trapped in the uninformative region.

## 2   The sample complexity of SGD

In this section, we focus on understanding the complexity of the SGD algorithm, i.e., how the number of total gradient steps $n$ scales with the dimension $d$ of the token embeddings, in the high-dimensional limit $d \gg 1$; for simplicity of exposition, we focus on the case $k = 1$, but the same arguments can be repeated for each of the components of the output.

**No positional encoding —**   We first focus on the case without positional encoding: $P = 0$ implies that the only relevant sufficient statistic is $m$. An analogy with single-index models for networks can be done: For single-index models, the *information exponent* fully characterizes the sample complexity of the SGD algorithm Ben Arous et al. [2021]. The definition can be generalized for sequential data.

**Definition 2** (Sequence Information Exponent (SIE)). *Given $\boldsymbol{f}_{\boldsymbol{w}_\star}^{\mathrm{SSI}}$ a sequence single-index model, let $g$ be the function that acts on the local field $\boldsymbol{z}_\star$, we define the* sequence information exponent *as*

$$SIE(\boldsymbol{f}_{\boldsymbol{w}_\star}^{\mathrm{SSI}}) := \min\left\{ \sum_{l=1}^{L} k_l > 0\colon \boldsymbol{k} \in \mathbb{N}^L, \mathbb{E}_{\boldsymbol{z}\sim\mathcal{N}(0,I_L)}\left[\left(\prod_{l=1}^{L}\mathrm{He}_{k_l}(z_l)\right)g(\boldsymbol{z})\right] \neq 0 \right\},$$

*where $\mathrm{He}_k$ is the $k$-th order Hermite polynomial.*

In Appendix B we provide more details on Hermite polynomials. Let us give some examples of SIE for different SSI models:

- $g(\boldsymbol{z}_\star) = z_{\star,1} + z_{\star,2} + \cdots + z_{\star,L}$ has SIE $= 1$;
- $g(\boldsymbol{z}_\star) = z_{\star,1}z_{\star,2} = \mathrm{He}_1(z_{\star,1})\mathrm{He}_1(z_{\star,2})$ has SIE $= 2$;
- $g(\boldsymbol{z}_\star) = \mathrm{He}_1(z_{\star,1})\mathrm{He}_4(z_{\star,2}) + \mathrm{He}_2(z_{\star,3})\mathrm{He}_2(z_{\star,4})$ has SIE $= 4$;
- $g(\boldsymbol{z}_\star) = \prod_{l=1}^{L}\mathrm{He}_{k_l}(z_{\star,l})$ has SIE $= \sum_{l=1}^{L}k_l$.

The main feature of the information exponent is that it can be connected to the sample complexity of the SGD algorithm; we prove an equivalent result for the sequence information exponent.

**Theorem 1** (Informal). *Let $\boldsymbol{f}_{\boldsymbol{w}_\star}^{\mathrm{SSI}}(X)$ be a sequence single-index model, and let SIE be its sequence information exponent. If the model $f_{\boldsymbol{w}}$ has a rich enough Hermite expansion, then the sample complexity of the SGD algorithm is*

$$t_\eta^+ = \begin{cases} \mathcal{O}_L(d) & \text{if SIE} = 1 \\ \mathcal{O}_L(d\log^2 d) & \text{if SIE} = 2 \\ \mathcal{O}_L(d^{SIE-1}) & \text{if SIE} \geq 3 \end{cases}.$$

A formal statement of the Theorem and the proof are given in the Appendix C. It relies on the following connection between the *flatness* of the landscape near initialization and the sequence information exponent:

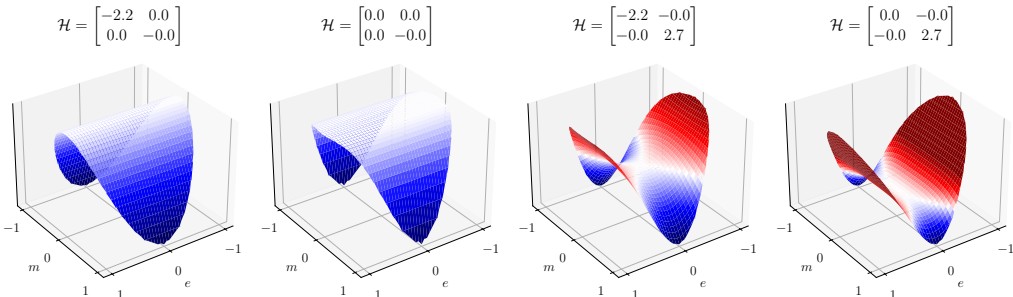

Figure 1: the landscape of the population risk, together with the hessian at initialization, for different values of the SIE and positional encoding. (left) $g(\boldsymbol{z}_\star) = \mathrm{He}_2(\boldsymbol{z}_{\star,1}) + \mathrm{He}_2(\boldsymbol{z}_{\star,2})$: SIE=2, no positional encoding: null gradient, but non-null hessian; (center-left) SIE=4, no positional encoding: the first non-null term at initialization is at the 4th order; (center-right) $g(\boldsymbol{z}_\star) = \mathrm{He}_4(\boldsymbol{z}_{\star,1}) + \mathrm{He}_4(\boldsymbol{z}_{\star,2})$: SIE=2, with positional encoding: again dynamic dominated by the hessian, but we have a positive curvature in the direction of $e$; (right) SIE=4, with positional encoding: hessian is positive semidefinite, and the dynamic is again at 4th order in direction of $e$. In all the examples $L = 2, P_1 = -P_2, R = \mathrm{Tr}$.

**Proposition 1.** *Define the* flatness *index* $\kappa(f_{\boldsymbol{w}}, \boldsymbol{f}_{\boldsymbol{w}_\star}^{\mathrm{SSI}})$ *of the model as*

$$\kappa(f_{\boldsymbol{w}}, \boldsymbol{f}_{\boldsymbol{w}_\star}^{\mathrm{SSI}}) = \min\left\{ k > 0 : \nabla^k R(\boldsymbol{0}, 0) \neq 0 \right\},$$

*where $R$ is the reduced population loss defined in* (9). *Then, if the model $f_{\boldsymbol{w}}$ has a rich enough Hermite expansion, then*

$$\kappa\left(f_{\boldsymbol{w}}, \boldsymbol{f}_{\boldsymbol{w}_\star}^{\mathrm{SSI}}\right) = \mathrm{SIE}\left(\boldsymbol{f}_{\boldsymbol{w}_\star}^{\mathrm{SSI}}\right)$$

Intuitively, a higher value of $\kappa$ implies that the landscape of $R$ is flatter around the initialization point $(\boldsymbol{0}, 0)$, and thus the number of gradient steps needed to build a weak correlation along either the $\boldsymbol{e}$ or $m$ directions is higher.

In Figure 1, we show some examples of the population loss landscape for different values of the SIE. Figure 1 focuses on even SIE because the symmetry of Equation (1) restricts the possible targets to even functions; in the Appendix D we discuss how to surpass this limitation, and we present settings with odd SIE.

**The effect of positional encoding —** Despite the fact that positional encoding only acts on the trained model, and not the target function, it changes the population loss, potentially changing the dynamic at initialization. In particular, adding positional encoding increase the expressivity of the model, and can ultimately lead to faster weak-recovery of the SGD.

**Lemma 1.** *Let $f_{\boldsymbol{w}}$ be a model with $P = 0$ that learns a target $\boldsymbol{f}_{\boldsymbol{w}_\star}^{\mathrm{SSI}}(X)$ with a given* SIE. *If we add a positional encoding $P$ to the model, and let $f_{\boldsymbol{w}}^{new}$ be new model, then*

$$\kappa\left(f_{\boldsymbol{w}}^{new}, \boldsymbol{f}_{\boldsymbol{w}_\star}^{\mathrm{SSI}}\right) \leq \kappa\left(f_{\boldsymbol{w}}, \boldsymbol{f}_{\boldsymbol{w}_\star}^{\mathrm{SSI}}\right) = \mathrm{SIE}\left(\boldsymbol{f}_{\boldsymbol{w}_\star}^{\mathrm{SSI}}\right).$$

In other words, the positional encoding can only decrease the flatness of the loss landscape, thus the sample complexity of the SGD algorithm could be reduced. The proof of this lemma is given in the Appendix C. In the right part of Figure 2, we present an example where adding the positional encoding can improve the sample complexity of the SGD algorithm. The left part of Figure 2 shows that the population loss landscape at initialization is flat, and the hessian is null: the first non-null term in the Taylor expansion is at order 4, hence the SIE is 4. The right part of Figure 2 shows instead that when we add

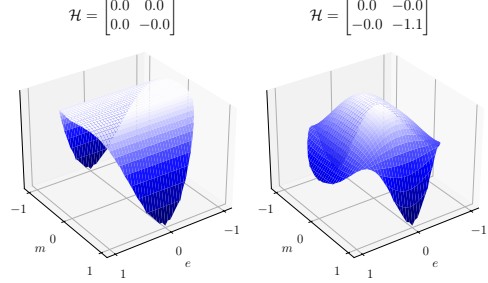

Figure 2: Population loss landscape for $P = 0$ (left) and $P \neq 0$ (right). Example of a case where SIE $= 4$, while $\mathrm{SIE}_{\mathrm{positional}} = 2$. Target: $g(\boldsymbol{z}_\star) = {}^4/_3 + \mathrm{He}_4(z_1) + 2\mathrm{He}_4(z_{\star,2}), P_1 = -P_2, R = \mathrm{Tr}.$

the positional encoding, a non-null term appears at order 2, and $\mathrm{SIE}_{\mathrm{positional}} = 2$. Note that while the positions of the global minima are not affected by the positional encoding, SGD can converge to the new local minima instead; more discussion on this point is given in Section 4. In contrast, the right part of Figure 1 shows that the positional encoding is not necessarily beneficial: there are cases for which the loss landscape changes, but not the sample complexity.

## 3    The role of the sequence length

In this section, we focus a first new emerging characteristic of the *sequence single-index models* over the vanilla *single-index models*: the sequence length $L$. Therefore, we neglect the effect of the positional encoding, by setting $P = 0$, in order to isolate just the effect of the sequence length. The goal of the section is to measure the speed-up that a model like Equation (1) can achieve over the plain *single-index models* when processing sequential data with length $L$.

**Linear attention —**    Our first goal is to understand the dependence of the convergence rate of SGD on the sequence length. For that, consider the particular case of *linear attention*, given by the reduction map:

$$R[A] = \boldsymbol{a}_{\mathrm{left}}^{\top} A \boldsymbol{a}_{\mathrm{right}} \quad \text{with } \boldsymbol{a}_{\mathrm{left}} = \boldsymbol{a}_{\mathrm{right}} = \frac{1}{\sqrt{L}} \left(1, 1, \dots, 1\right)^{\top} \in \mathbb{R}^L. \tag{12}$$

Rearranging the terms

$$f_{\boldsymbol{w}}(\boldsymbol{z}) = \boldsymbol{a}_{\mathrm{left}}^{\top} \left(\boldsymbol{z}\boldsymbol{z}^{\top}\right) \boldsymbol{a}_{\mathrm{right}} = \frac{1}{L} \sum_{i=1}^{L} \sum_{j=1}^{L} z_i z_j = \left(\sum_{i=1}^{L} \frac{z_i}{\sqrt{L}}\right)^2 = \left(\frac{\mathrm{flatten}(X) \cdot \boldsymbol{w}_{\mathrm{tied}}}{\sqrt{L}}\right)^2, \tag{13}$$

where $\boldsymbol{w}_{\mathrm{tied}} := \mathrm{concat}(\boldsymbol{w}, \dots, \boldsymbol{w}) \in \mathbb{R}^{Ld}$ is the concatenation of $L$ copies of $\boldsymbol{w}$. This model is equivalent to a *generalized linear model* with *tied weights*, and activation function $\sigma(x) = x^2$. In terms of performance, taking a general activation $\sigma$ will at worst be the same of the *attention mechanism* originally considered, if not better. More precisely, we consider:

$$f_{\boldsymbol{w}}(X) = \sigma \left(\frac{\mathrm{flatten}(X) \cdot \boldsymbol{w}_{\mathrm{tied}}}{\sqrt{L}}\right). \tag{14}$$

This is the most generic model we study in this section. Further numerical experiments elucidating the equivalence of the speed-up for this *tied network* and for the attention models can be found in Appendix E. Note that *tied networks* are not restricted to learn even function only, differently from the model in Equation (1).

**The corresponding untied network —**    Given the model in Equation (14), a natural benchmark is the model with untied weights. Let $W \in \mathbb{R}^{L \times d}$ be the matrix of weights, whose rows are all updated with Equation (2), the untied network is given by

$$f_W(X) = \sigma \left(\frac{\mathrm{flatten}(X) \cdot \mathrm{flatten}(W)}{\sqrt{L}}\right). \tag{15}$$

Since we have $L$ independent weights (the rows of $W$) the sufficient statistic measuring the overlap between the model and the target SSI is not a scalar as for the tied network, but a vector of length $L$

$$\boldsymbol{m} = \frac{W \boldsymbol{w}_{\star}}{d} \in \mathbb{R}^L \quad \text{compacted to a scalar as} \quad m_{\mathrm{untied}} = \frac{\|\boldsymbol{m}\|}{\sqrt{L}}. \tag{16}$$

**Measuring the speedup —**    The learning rate plays an important role in determining the number of gradient steps needed to reach weak recovery: the larger the learning rate $\gamma(L)$ is, the faster the model learns. However, if it becomes too large, SGD will fail to converge, never achieving weak recovery. The gradient-flow approximation in Eq. (11) exhibits this effect: when $\gamma$ becomes too large the dynamic of the system is not attracted by $\boldsymbol{w}_{\star}$ or $P$ anymore, and there is no learning. In order to have a faithful measure of the speed-up, we will assume that the learning rate is taken to be the largest possible that guarantees weak recovery; we discuss this upper bound on the learning rate in App. E.

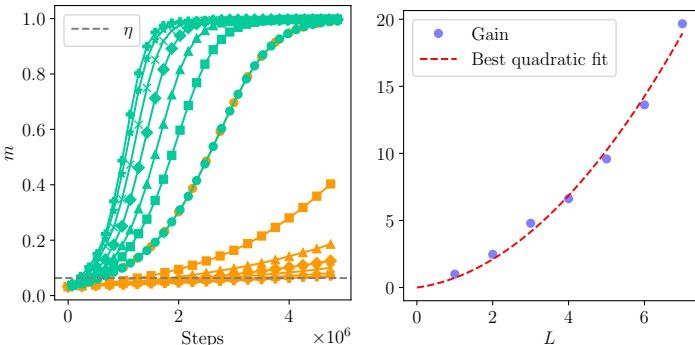

Figure 3: Left: overlap $m$ for tied (green) and untied (orange) networks as a function of the number of gradient steps; different symbols represent different values of $L$. Right: measured gain as a function of the sequence length $L$, with the best fit line showing its scaling as $L^2$. $g(\boldsymbol{z}_\star) = \sum_{i=1}^{L} \mathrm{He}_2(z_{\star,i})$, $d = 1000$, $\sigma = \mathrm{ReLU}$.

We measure the speed-up of tied networks with respect to untied networks, as measured in terms of number of gradient steps needed to reach weak recovery, by the ratio of the weak recovery times in the two cases

$$\mathrm{gain}(L) := \frac{\tau_{\eta,\mathrm{untied}}^{\mathrm{weak}}}{\tau_\eta^{\mathrm{weak}}}. \tag{17}$$

where $\tau_{\eta,\mathrm{untied}}^{\mathrm{weak}}$ is given by Definition 1 where $m$ is replaced by $m_{\mathrm{untied}}$; the dependence of gain on $\eta$ is subleading, and we will neglect it in the following.

**Theorem 2.** *Let $C_{\mathrm{SIE}} \in \mathbb{R}^{L^{\mathrm{SIE}}}$ be the first non-zero tensor in the Hermite expansion of $g$ (see Appendix B). Then the gain satisfies with high probability*

$$\mathrm{gain} \gtrsim \left( \frac{C_{\mathrm{SIE}} \times (\mathbf{1}, \dots, \mathbf{1})}{\|C_{\mathrm{SIE}}\|_{\mathrm{op}}} \right)^2 \cdot \begin{cases} L & \textit{if } \mathrm{SIE} = 1 \\ 1 & \textit{otherwise} \end{cases}.$$

*If the tensor $C_{\mathrm{SIE}}$ is* orthogonally decomposable*, in particular in the cases where $\mathrm{SIE} \le 2$ or $g$ is separable, then*

$$\mathrm{gain} \asymp \left( \frac{C_{\mathrm{SIE}} \times (\mathbf{1}, \dots, \mathbf{1})}{\|C_{\mathrm{SIE}}\|_{\mathrm{op}}} \right)^2 \cdot \begin{cases} L & \textit{if } \mathrm{SIE} = 1 \\ 1 & \textit{otherwise} \end{cases}.$$

By definition of the operator norm, we have

$$C_{\mathrm{SIE}} \times (\mathbf{1}, \dots, \mathbf{1}) \le \|C_{\mathrm{SIE}}\|_{\mathrm{op}} L^{\mathrm{SIE}/2}, \quad \text{and hence} \quad 0 \le \mathrm{gain} \lesssim L^{\mathrm{SIE} \vee 2}.$$

Since the untied network has $L$ times the number of parameters compared to the tied one, a naive parameter counting argument would yield a $\mathrm{gain} = L^{(SIE-1)\vee 1}$ expected gain. Counter-intuitively, the actual gain of using a tied network can either exceed or fall short of this naive value, depending on the function $g$. In pathological cases (see Appendix E), the tied network can even either fail to learn the target, or do so slower than its untied counterpart.

**Example for** $\mathrm{SIE} = 2$ — Let's assume to have a target function $g(\boldsymbol{z}_\star) = \sum_{i=1}^{L} \mathrm{He}_2(z_{\star,i})$. In this case, the SIE is 2 and the tensor $C_2$ is simply the identity matrix $I_L$. The gain is by

$$\mathrm{gain} \asymp \left( \frac{I_L \times (\mathbf{1}, \dots, \mathbf{1})}{\|I_L\|_{\mathrm{op}}} \right)^2 \cdot 1 = \left( \frac{L}{1} \right)^2 \cdot 1 = L^2.$$

Fig. 3 show a numerical experiment, with a ReLU activation, confirming the result of Th. 2.

## 4 Positional encoding and training dynamics

We now turn our attention to the role played by positional encoding in the attention layer when trained under SGD. Since the focus is on the effect of positional encoding, we stick with a class of target

functions that can exhibit either a *semantic* (label mostly depends on tokens value, but not the order) or a *positional* (where tokens most important feature is their position in the sequence, rather then their embedding). Consider a SSI target function of the form

$$\boldsymbol{f}_{\boldsymbol{w}_\star}^{\mathrm{SSI}}(X) = (1-\omega)\operatorname{softmax}\left(X\boldsymbol{w}_\star\boldsymbol{w}_\star^\top X^\top\right) + \omega \operatorname{softmax}\left[\begin{pmatrix} a^2 & -a^2 \\ -a^2 & a^2 \end{pmatrix}\right] \in \mathbb{R}^{2\times 2}, \quad (18)$$

where he parameter $\omega \in [0,1]$ allows the target to switch from a semantic to a positional behavior, while the parameter $a \in (0,1]$ controls the alignment of the target with its positional part.

We shall train the model in Equation (1) with positional encoding $P_1 = -P_2$ and reduction map $R$ the identity function. We focus on the *gradient flow* limit where $\eta$ is sufficiently small [Robbins and Monro, 1951]. Our analysis will focus on the population loss given by Equation (9), since it completely characterize the behavior of SGD in this regime; the sufficient statistics $(\boldsymbol{e}, m)$ are the only free variables of the setting.

The sequential information exponent of this setting is SIE $= 2$ (see Appendix F for the explicit derivation), thus the sample complexity for escaping the initialization is $\mathcal{O}(d \log d)$. After the initial phase, SGD fast converges to the minimum of the population loss that is fully aligned with either the semantic or the positional sufficient statistic, i.e. $\big\|(\boldsymbol{e}, m)\big\| = 1$, but is not guaranteed to be the global minimum. Figure 5 shows an example where the population loss has 2 minimums, one semantic with $(e, m) = (1, 0)$ and one positional with $(e, m) = (0, 1)$, and the steepest direction at initialization, namely the eigenvector associated with the lowest eigenvalue of the Hessian, points towards the local one; in this case, the gradient flow will converge to the wrong minima.

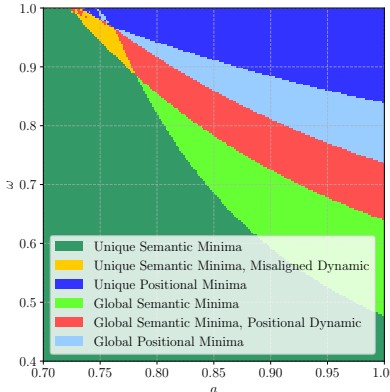

Figure 4: Different behaviors of SGD depending on the parameters $\omega$ and $a$.

Varying the parameters, $\omega$ and $a$ the high-dimensional SGD dynamics from random initialization exhibits a rich phenomenology. In Figure 4 we show the phase diagram with all the possible behaviors:

- **Unique Positional Minima**: the population loss has unique positional minima, and the SGD converges to it. This is the case for $\omega = 1$ and $a = 1$.

- **Global Positional Minima**: the population loss has both a semantic and a positional minimum, and SGD converges to the global positional one.

- **Global Semantic Minima, Positional Dynamic**: the population loss has both a semantic and a positional minimum, and SGD converges to the local positional one. This is the case where SGD **does not converge** to the global minima.

- **Unique Semantic Minima, Misaligned Dynamics**: the population loss has unique semantic minima, and the SGD converges to it, even though the steepest direction at initialization points orthogonal to it.

- **Global Semantic Minima**: the population loss has both a semantic and positional minima, and SGD converges to the global semantic one.

- **Unique Semantic Minima**: the population loss has unique semantic minima, and the SGD converges to it. This is the case for $\omega = 0$ and $a = 1$.

We verify this by simulating many runs of SGD with different initializations and data samples. In Figure 5 we compute the empirical probability of convergence to the semantic minima, for $a = 1$ and varying $\omega$. The theoretical value of $\omega$ where we have a transition from sematic to positional dynamics is $\omega_{\mathrm{trans}} = 0.64$, which is in good agreement with the transition observed in the measured probabilities; the transition becomes sharper as $d$ increases: ideally, in the limit $d \to \infty$ we expect a step function. The simulations in Figure 5 are performed with $d = 1000$, and some finite size effects are still present. In Appendix F we show present a more detailed analysis, including different values of $d$.

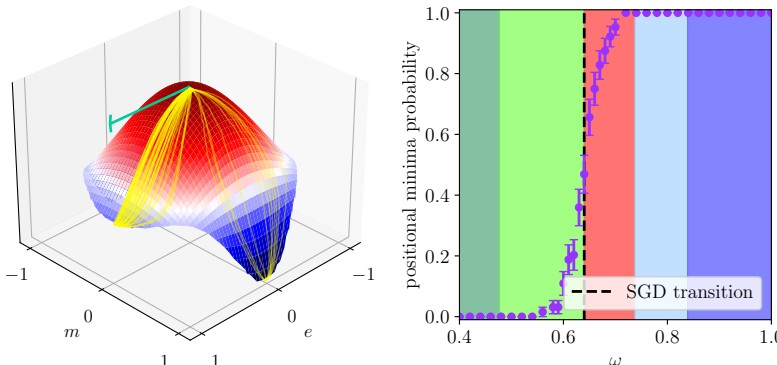

Figure 5: (left) surface of the population loss for $\omega = 0.67$ and $a = 1$. The steepest direction at initialization (green vector) points towards the *positional* local minimum, while the global minima is semantic. Some examples of SGD trajectories are shown in yellow: most of them fall into the semantic local minimum, while some others manage to fully-recover the global minimum due to finite size effects ($d = 1000$). (right) empirical probability of convergence to the semantic minima as a function of $\omega$ for $a = 1$. The probability is computed over 64 SGD runs with different initializations and data samples. The theoretical prediction of the transition from semantic to positional minima is at $\omega_{\mathrm{trans}} \approx 0.64$.

## Conclusion

In this paper, we introduced the Sequence-Single Index model as a new, high-dimensional theoretical framework for analyzing single-layer attention architectures. The most significant contribution of this study is the definition of the Sequence Information Exponent. This exponent, which serves as a rigorous tool to quantify the inherent hardness and predict the sample complexity of Stochastic Gradient Descent, is defined in direct analogy with classical single-index models. This analysis transcends qualitative understanding, providing precise scaling laws for the required number of gradient steps, denoted by $n$, relative to the token dimension, denoted by $d$. We demonstrated that the sequential setting is significantly richer, showing how the sequence length $L$ accelerates convergence and how positional encoding can proactively reduce the SIE, thereby lowering the computational barrier to learning.

This work establishes a foundational line of research essential for a principled understanding of modern sequential data models, particularly those based on the Transformer architecture. The intricate dynamics manifesting in the population loss landscape, exemplified by the identification of phase transitions and the convergence to suboptimal local minima, unveil a wealth of avenues for future investigation. Subsequent research should aim to fully map these complex high-dimensional dynamics, develop techniques for robustly breaking inherent symmetries, and extend the SIE framework to more complex multi-index and multi-layer attention systems. We hope that this quantitative framework will stimulate further theoretical explorations, leading to the development of a robust, generalizable understanding of learning on structured sequential data.

**Limitations –** The theoretical analysis of SGD in Sequence Single-Index models is subject to several simplifications for tractability. Specifically, the model under scrutiny is a simplified, single-layer attention architecture where the key and query matrices are tied, the attention head dimension is one ($d_{head} = 1$), and the value matrix is the identity ($V = I_L$). It should be noted that the scope of this architecture is restricted to a specific class of target functions, including even functions, although Appendix D provides an extension to this class. Additionally, the analysis operates under the assumption of token independence across the sequence, a crucial simplification that facilitates the execution of numerous sequence modeling tasks. The analysis of SGD dynamics is founded on the approximation of the discrete dynamics via a second-order ODE (Gradient Flow approximation).

**Acknowledgement—** We would like to thank Luca Pesce, Luca Biggio, and Yatin Dandi for their insightful discussions. We acknowledge funding from the Swiss National Science Foundation grants SNSF SMArtNet (grant number 212049), OperaGOST (grant number 200021 200390), DSGIANGO (grant number 225837) and by the French government, managed by the National Research Agency (ANR), under the France 2030 program with the reference "ANR-23-IACL-0008" and the Choose France - CNRS AI Rising Talents program.

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

# A  Reduction from attention to sequence single-index

Let $X \in \mathbb{R}^{L \times d}$ denote a sequence of length $L$ of $d$-dimensional tokens, and consider the standard dot-product attention:

$$\text{Attention}(X) = \text{softmax}\left(\frac{QK^\top}{\sqrt{d_{\text{head}}}}\right)V \tag{19}$$

where $Q = (X + P)W_Q, K = (X + P)W_K, V = (X + P)W_V \in \mathbb{R}^{L \times d_{\text{head}}}$ are trainable weights known as the *query*, *key* and *value* matrices, respectively. The matrix $P \in \mathbb{R}^{L \times d}$ is the *positional encoding*, a fixed matrix needed to inject a representation of the position of the tokens in the sequence. To make the analysis tractable, [Troiani et al., 2025] considered the following simplifying assumptions:

- Key and query matrix are tied, and $d_{\text{head}} = 1$: $Q = K = X \cdot \boldsymbol{w} \in \mathbb{R}^{L \times 1}$;
- Identity value matrix $V = I_L$.

Note the second assumption is mild for single-layer attention, since we do not need to learn a new representation of the sequence. Under these assumptions, eq. (19) reduces to:

$$\text{TiedAttention}(X) = \text{softmax}\left(X\boldsymbol{w}\boldsymbol{w}^\top X^\top\right) \tag{20}$$

This is a map from sequences $X \in \mathbb{R}^{L \times d}$ to $L \times L$ matrices. Further adding the reduction map $R : \mathbb{R}^{L \times L} \to \mathbb{R}^k$, we get the model in eq. (1). Finally, to get the reduction to a sequence single-index model, it suffices to consider $P = 0$ and the map on real-valued sequences $\boldsymbol{s} \in \mathbb{R}^L$:

$$g(\boldsymbol{s}) = R\left[\text{softmax}\left(\boldsymbol{s}\boldsymbol{s}^\top\right)\right] \tag{21}$$

# B  Mathematical preliminaries and notations

## B.1  Tensors

We consider tensors as multidimensional arrays: a tensor $T$ of order $k$ and dimensions $(d_1, \ldots, d_k)$ is simply an element of $\mathbb{R}^{d_1 \times \cdots \times d_k}$. Its elements are denoted by $T_{i_1 \ldots i_k}$, where $i_\ell \in [d_\ell]$. The scalar product between two tensors with same dimensions is defined as

$$\langle T, T \rangle = \sum_{i_1, \ldots, i_k} T_{i_1 \ldots i_k} T'_{i_1 \ldots i_k}.$$

We say that a tensor is *symmetric* if all its dimensions are equal and for any index $(i_1, \ldots, i_k)$ and permutation $\sigma \in \mathfrak{S}_k$,

$$T_{i_1 \ldots i_k} = T_{i_{\sigma(1)} \ldots i_{\sigma(k)}}.$$

We shall need two operations on tensors: the first is the *tensor product*, that turns two tensors of order $k, \ell$ into a tensor of order $k + \ell$ defined as

$$(T \otimes T')_{i_1 \ldots i_{k+\ell}} = T_{i_1 \ldots i_k} T'_{i_{k+1} \ldots i_{k+\ell}}.$$

The second is the *tensor-matrix* contraction: given a tensor $T$ of order $k$, an index $\ell$ and a matrix $M$ of size $d_\ell \times d'\ell$, the tensor $T \times_\ell M$ is defined as

$$(T \times_\ell M)_{i_1 \ldots i'_\ell \ldots i_k} = \sum_{i_\ell} T_{i_1 \ldots i_\ell \ldots i_k} M_{i_\ell i'_\ell}$$

Given $k$ matrices $M^{(1)}, \ldots, M^{(k)}$, we will use the shorthand

$$T \times (M^{(1)}, \ldots, M^{(k)}) = T \times_1 M^{(1)} \cdots \times_k M^{(k)}$$

Immediate properties of those operations are gathered in the following lemma:

**Lemma 2.** *The operation $\times$ is associative: if $T$ is a tensor and $(M^{(1)}, \ldots, M^{(k)}), (N^{(1)}, \ldots, N^{(k)})$ are matrices with compatible dimensions,*

$$\left(T \times (M^{(1)}, \ldots, M^{(k)})\right) \times (N^{(1)}, \ldots, N^{(k)}) = T \times (M^{(1)}N^{(1)}, \ldots, M^{(k)}M^{(k)})$$

*Let $T \in \mathbb{R}^{d_1 \times \cdots \times d_k}$ and $(\boldsymbol{x}_1, \ldots, \boldsymbol{x}_k) \in \mathbb{R}^{d_1} \times \cdots \times \mathbb{R}^{d_k}$. Then*

$$\langle T, \boldsymbol{x}_1 \otimes \cdots \otimes \boldsymbol{x}_k \rangle = T \times (\boldsymbol{x}_1, \ldots, \boldsymbol{x}_k).$$

**Odeco tensors** Since tensors of order $k \geq 3$ are sometimes hard to handle, we work with a restricted class. We say that a symmetric tensor $T$ is *odeco* (short for *orthogonally decomposable*, see Robeva [2016]) if there exist real numbers $\lambda_1, \ldots, \lambda_r$ and orthogonal vectors $\boldsymbol{v}_1, \ldots, \boldsymbol{v}_r$ such that

$$T = \sum_{i=1}^{r} \lambda_i v_i^{\otimes k}.$$

In particular, all tensors of order 1 (with $r = 1$) and 2 (with $r$ equal to the rank of $T$) are odeco.

## B.2 Hermite Polynomials

In this section we provide our definition of Hermite polynomials, which are used in the construction of the Hermite basis, both for the one-dimensional and the multidimensional case.

**Gaussian measure and Gaussian $\ell^2$ space** We define the Gaussian density in $p$ dimensions

$$\omega_p(\boldsymbol{x}) = \frac{1}{(2\pi)^{d/2}} \exp\left(-\frac{\|\boldsymbol{x}\|^2}{2}\right),$$

and $\mathrm{d}\omega_p(\boldsymbol{x}) = \omega_p(\boldsymbol{x})\mathrm{d}\boldsymbol{x}$. This measure defines a space $\ell^2(\omega_p)$ of functions $f$ satisfying

$$\|f\|_{\omega} := \int f(\boldsymbol{x})^2 \mathrm{d}\omega_p(\boldsymbol{x}) < \infty;$$

it is a Hilbert space w.r.t the scalar product

$$\langle f, g \rangle_{\omega} = \int f(\boldsymbol{x})g(\boldsymbol{x})\mathrm{d}\omega_p(\boldsymbol{x}).$$

**Hermite polynomials and tensors** We follow the conventions of Grad [1949]. Define the $k$-th Hermite tensor $\mathcal{H}_k$ as

$$\mathcal{H}_k = \frac{(-1)^k}{\omega_p} \nabla^k \omega_p,$$

where $\nabla^k$ is the $k$-th order derivative. This results in a $k$-th order symmetric tensor of size $p \times \cdots \times p$. The Hermite tensors are orthogonal, in the sense that

$$\langle (\mathcal{H}_k)_{i_1 \ldots i_k}, (\mathcal{H}_\ell)_{j_1 \ldots j_\ell} \rangle_{\omega} \neq 0 \quad \text{if and only if} \quad k = \ell \text{ and } (i_1, \ldots, i_k) \text{ is a permutation of } (j_1, \ldots, j_\ell).$$

When $p = 1$, all Hermite tensors are scalars, and we get the usual Hermite polynomials:

$$\mathrm{He}_0(x) = 1, \tag{22}$$
$$\mathrm{He}_1(x) = x, \tag{23}$$
$$\mathrm{He}_2(x) = x^2 - 1, \tag{24}$$
$$\mathrm{He}_3(x) = x^3 - 3x, \tag{25}$$
$$\mathrm{He}_4(x) = x^4 - 6x^2 + 3. \tag{26}$$

**Hermite expansion** The orthogonality properties of the Hermite tensors imply the following theorem:

**Theorem 3.** *Let $f \in \ell^2(\omega_p)$. There exist a unique sequence of coefficients $\left(C_k(f)\right)_{k \geq 0}$ such that $C_k(f)$ is a tensor of order $k$ and*

$$f = \sum_{k \geq 0} \langle C_k(f), \mathcal{H}_k \rangle. \tag{27}$$

*Those coefficients are given by the following formula:*

$$C_k(f) = \frac{1}{k!} \int f(\boldsymbol{x}) \mathcal{H}_k(\boldsymbol{x}) \mathrm{d}\omega_p(\boldsymbol{x}).$$

*Further, the scalar product $\langle \cdot, \cdot \rangle_{\omega}$ can be written as*

$$\langle f, g \rangle_{\omega} = \sum_{k \geq 0} \frac{1}{k!} \langle C_k(f), C_k(g) \rangle$$

The proof of this theorem can be found in Grad [1949]. The identity (27) is called the *Hermite expansion* of $f$, and the $C_k(f)$ are its *Hermite coefficients*.

Finally, by invariance of the Gaussian distribution through orthogonal transformation, the following holds:

**Lemma 3.** *Let $g : \mathbb{R}^p \to \mathbb{R}$, and $W \in \mathbb{R}^{p \times q}$ be a matrix satisfying $WW^\top = I_p$. Let $f(\boldsymbol{x}) = g(W\boldsymbol{x})$. Then*

$$C_k(f) = C_k(g) \times (W, \dots, W).$$

When $p = 1$ and $\boldsymbol{w}$ is a single vector, we get

$$C_k(f) = c_k(g)\boldsymbol{w}^{\otimes k}.$$

This gives rise to a link between odeco tensors and separable functions:

**Lemma 4.** *Let $g : \mathbb{R}^\ell \to \mathbb{R}$ be a separable function, such that*

$$g(\boldsymbol{z}) = \sum_i g_i(z_i),$$

$W \in \mathbb{R}^{\ell \times q}$ *an orthogonal matrix, and let $f(\boldsymbol{x}) = g(W\boldsymbol{x})$. Then*

$$C_k(f) = \sum_{i=1}^{\ell} c_k(g_i)\boldsymbol{w}_i^{\otimes k},$$

*and in particular every Hermite coefficient of $f$ is odeco.*

## C  Formalization and proofs

### C.1  Preliminaries

We consider the following approximation of the SGD dynamics:

$$\frac{d\boldsymbol{w}}{dt} = -\mathbb{E}\Big[\nabla_{\boldsymbol{w}}^\perp \mathcal{L}(X, y, f_{\boldsymbol{w}})\Big] - \gamma \mathbb{E}\Big[\big\|\nabla_{\boldsymbol{w}}^\perp \mathcal{L}(X, y, f_{\boldsymbol{w}})\big\|^2\Big]\boldsymbol{w} \tag{28}$$

The results of Ben Arous et al. [2021] (when $m$ is a scalar) and Arnaboldi et al. [2024c] (when $\boldsymbol{m}$ is a vector) imply the following:

**Theorem 4.** *Let $\tau_\eta$ be the weak recovery time of the ODE (28), with the same initial conditions as the process (2). Then for small enough $\eta$ and any $\delta > 0$, there exist constants $c(\delta), C(\delta)$ such that if*

$$\gamma = c(\delta)(d\tau_\eta)^{-1}, \tag{29}$$

*then with probability at least $1 - \delta$*

$$t_\eta^+ \leq C(\delta)d\tau_\eta^2.$$

*On the other hand, for any $t \leq C(\delta)d\tau_\eta^2$, if $\gamma \leq c(\delta)(dt)^{-1/2}$, then with probability $1 - \delta$*

$$t_\eta^+ \geq c(\delta)t$$

When $\gamma$ does not satisfy the bound (29), we cannot show a strong enough concentration around the deterministic ODE dynamics, and hence directly showing non-convergence of (2) is difficult. However, as we shall see in the proof, above this value of $\gamma$ the inhibitive term in (28) dominates at initialization, and hence the ODE dynamics stay trapped around zero overlap with the target subspace. For this reason, we shall consider (in line with Ben Arous et al. [2021]) that the sample complexity cannot be improved by increasing $\gamma$ above the bound (29).

**Remark.** *When $\gamma$ is instead fixed below the value (29), the hitting time of the dynamics 2 is instead given by*

$$t_\eta^+(\gamma) \asymp \gamma^{-1}\tau_\eta.$$

We shall also assume that $m_0 > 0$, and that the coefficients appearing in the gain expression in Theorem 2 are all non-negative. As mentioned in Ben Arous et al. [2021], Arnaboldi et al. [2024b,c], this condition can be ensured with probability $1/2$ by randomly setting the learning rate to $\pm\gamma$ with equal probability.

## C.2 An ODE for overlap evolution

We begin by computing the expectation of the gradient term:

**Lemma 5.** *In the tied case, we have when $\|\boldsymbol{w}\| = 1$*

$$\mathbb{E}\big[\mathcal{L}(X, y, f_{tied})\big] = \mathbb{E}[y^2] - 2\sum_{k \geq 0} c_k(\sigma)C_k(g) \times (\mathbf{1}_L, \dots, \mathbf{1}_L)m^k + \|\sigma\|_\omega.$$

*In the untied case, we have instead*

$$\mathbb{E}\big[\mathcal{L}(X, y, f_{untied})\big] = \mathbb{E}[y^2] - 2\sum_{k \geq 0} c_k(\sigma)C_k(g) \times (\boldsymbol{m}, \dots, \boldsymbol{m}) + \|\sigma\|_\omega.$$

*Proof.* Recall that $\mathcal{L}(X, y, f) = (y - f(X))^2 = y^2 - 2yf(X) + f(X)^2$. For simplicity, define

$$\tilde{\boldsymbol{w}}_{\text{tied}} = (\boldsymbol{w}\,\boldsymbol{w}\dots\boldsymbol{w}) \in \mathbb{R}^{dL}, \quad \tilde{\boldsymbol{w}}_{\text{untied}} = (\boldsymbol{w}_1\dots\boldsymbol{w}_L) \in \mathbb{R}^{dL} \quad \text{and} \quad \tilde{W}^\star = \begin{pmatrix} \boldsymbol{w}_1^\star & \mathbf{0} & \cdots & \mathbf{0} \\ \mathbf{0} & \boldsymbol{w}_2^\star & \cdots & \mathbf{0} \\ \vdots & \vdots & \ddots & \vdots \\ \mathbf{0} & \mathbf{0} & \cdots & \boldsymbol{w}_L^\star \end{pmatrix} \in \mathbb{R}^{L \times dL}$$

Then, for $\square \in \{\text{tied}, \text{untied}\}$, we have

$$f_\square(X) = \sigma\left(\frac{\langle \tilde{\boldsymbol{w}}_\square, \text{flatten}(X)\rangle}{\sqrt{L}}\right) \quad \text{and} \quad y(X) = g(\tilde{W}^\star \cdot \text{flatten}(X)).$$

Since $\frac{\text{flatten}(X)}{\sqrt{L}}$ is a standard normal vector, and both $\frac{\tilde{\boldsymbol{w}}_\square}{\sqrt{L}}$ and $\frac{\tilde{W}^\star}{\sqrt{L}}$ are orthogonal matrices, we can use the Hermite expansion properties to find

$$\begin{aligned}
\mathbb{E}\big[f_\square(X)y(X)\big] &= \sum_{k \geq 0}\langle C_k(f_\square), C_k(y)\rangle \\
&= \sum_{k \geq 0}\langle c_k(\sigma)\tilde{\boldsymbol{w}}_\square^{\otimes k}, C_k(g) \times (\tilde{W}^\star, \dots, W^\star)\rangle \\
&= \sum_{k \geq 0} c_k(\sigma)C_k(g) \times (\tilde{W}^\star\tilde{\boldsymbol{w}}_\square, \dots, \tilde{W}^\star\tilde{\boldsymbol{w}}_\square).
\end{aligned}$$

In the tied case, we have $\tilde{W}^\star\tilde{\boldsymbol{w}}_{\text{tied}} = m\mathbf{1}_L$, while in the untied case $\tilde{W}^\star\tilde{\boldsymbol{w}}_{\text{untied}} = \boldsymbol{m}$. For the last term, we have in both cases $\frac{\langle \tilde{\boldsymbol{w}}_\square, \text{flatten}(X)\rangle}{\sqrt{L}} \sim \mathcal{N}(0, 1)$ whenever $\|\tilde{\boldsymbol{w}}_\square\|^2 = L$, and hence

$$\mathbb{E}[f_\square(X)^2] = \|\sigma\|_\omega.$$

This ends the proof. $\qquad\square$

When $\|\boldsymbol{w}\|$ (resp. $\|\boldsymbol{w}_i\|$ is different from one, the expressions of Lemma 5 depend on $q = \|\boldsymbol{w}\|^2$ (resp. $q_i = \|\boldsymbol{w}_i\|^2$). However, we can write

$$\nabla_{\boldsymbol{w}}\mathbb{E}[\mathcal{L}(X, y, f_{\text{tied}})] = \frac{\partial}{\partial m}\mathbb{E}[\mathcal{L}(X, y, f_{\text{tied}})]\boldsymbol{w}^\star + 2\frac{\partial}{\partial q}\mathbb{E}[\mathcal{L}(X, y, f_{\text{tied}})]\boldsymbol{w}.$$

As a result, we have

$$\nabla_{\boldsymbol{w}}^\perp\mathbb{E}[\mathcal{L}(X, y, f_{\text{tied}})] = \left(\frac{\partial}{\partial m}\mathbb{E}[\mathcal{L}(X, y, f_{\text{tied}})]\right)(\boldsymbol{w}^\star - m\boldsymbol{w}).$$

The same holds for the untied case:

$$\nabla_{\boldsymbol{w}}^\perp\mathbb{E}[\mathcal{L}(X, y, f_{\text{tied}})] = \left(\frac{\partial}{\partial m_i}\mathbb{E}[\mathcal{L}(X, y, f_{\text{tied}})]\right)(\boldsymbol{w}^\star - m_i\boldsymbol{w}_i)$$

We arrive at the following result:

**Proposition 2.** *Define the drift functions*

$$\phi_{tied}(m) = 2(1 - m^2) \sum_{k \geq 0} k c_k(\sigma) C_k(g) \times (\mathbf{1}_L, \ldots, \mathbf{1}_L) m^{k-1}$$

$$\phi_{untied}(\boldsymbol{m}) = 2(1 - \boldsymbol{m} \circ \boldsymbol{m}) \circ \sum_{k \geq 0} c_k(\sigma) C_k(g) \times (I_L, \boldsymbol{m}, \ldots, \boldsymbol{m}).$$

*Then the tied and untied overlaps satisfy the following ODEs:*

$$\frac{dm}{dt} = \phi_{tied}(m) - \gamma \mathbb{E}\left[\left\|\nabla_{\boldsymbol{w}}^{\perp} \mathcal{L}(X, y, f_{\boldsymbol{w}})\right\|^2\right] m \tag{30}$$

$$\frac{dm}{dt} = \phi_{untied}(\boldsymbol{m}) - \gamma \mathbb{E}\left[\left\|\nabla_{\boldsymbol{w}}^{\perp} \mathcal{L}(X, y, f_{\boldsymbol{w}})\right\|^2\right] \boldsymbol{m} \tag{31}$$

### C.3   Controlling the gradient norm

We turn our attention to the inhibitive terms in Proposition 2. We show the following:

**Lemma 6.** *There exist an $\eta > 0$ and two constants $c, C$ such that if $m < \eta$ (resp. $\|m\| \leq \eta$),*

$$cd \leq \mathbb{E}\left[\left\|\nabla_{\boldsymbol{w}}^{\perp} \mathcal{L}(X, y, f_{\boldsymbol{w}})\right\|^2\right] \leq Cd$$

*Proof.* We only treat the tied case; the untied one is done similarly. Differentiating the loss w.r.t $\boldsymbol{w}$, we find

$$\nabla_{\boldsymbol{w}} \mathcal{L}(X, y, f) = 2\sigma'\left(\frac{\langle \boldsymbol{w}_{\text{tied}}, \text{flatten}(X) \rangle}{\sqrt{L}}\right)\left(\sigma'\left(\frac{\langle \boldsymbol{w}_{\text{tied}}, \text{flatten}(X) \rangle}{\sqrt{L}}\right) - y(X)\right) \cdot \frac{\sum_i \boldsymbol{x}_i}{\sqrt{L}}$$

$\square$

We write $\boldsymbol{x}_i = \boldsymbol{x}_i^{\parallel} + \boldsymbol{x}_i^{\perp}$, where $\boldsymbol{x}_i^{\parallel}$ is the projection on $\boldsymbol{x}_i$ on the subspace spanned by $\boldsymbol{w}$ and $\boldsymbol{w}^{\star}$. Then

$$\nabla_{\boldsymbol{w}}^{\perp} \mathcal{L}(X, y, f) = 2\sigma'\left(\frac{\langle \boldsymbol{w}_{\text{tied}}, \text{flatten}(X) \rangle}{\sqrt{L}}\right)\left(\sigma'\left(\frac{\langle \boldsymbol{w}_{\text{tied}}, \text{flatten}(X) \rangle}{\sqrt{L}}\right) - y(X)\right) \cdot \frac{\sum_i (I - \boldsymbol{w}\boldsymbol{w}^{\perp}) \boldsymbol{x}_i^{\parallel} + \sum_i \boldsymbol{x}_i^{\perp}}{\sqrt{L}}$$

Importantly, the vector $x_i^{\perp}$ is independent from any of the prefactors, and has norm $d - 2$, while the first vector is a fixed-dimensional Gaussian. As a result, we have

$$\mathbb{E}\left[\left\|\nabla_{\boldsymbol{w}}^{\perp} \mathcal{L}(X, y, f)\right\|^2\right] = \mathbb{E}\left[f_{\boldsymbol{w}}(X)^2 (y(X) - f_{\boldsymbol{w}}(X))^2\right](d - 2) + O(1)$$

It remains to show that the expectation above is bounded away from zero. Assuming that the labels are centered for simplicity, when $m = 0$ the expectation simplifies to

$$L_0 = \mathbb{E}\left[f_{\boldsymbol{w}}(X)^2\right] \mathbb{E}\left[y(X)^2\right] + \mathbb{E}\left[f_{\boldsymbol{w}}(X)^4\right] > 0.$$

By continuity, we can choose $\eta > 0$ such that if $|m| \leq \eta$

$$\frac{L_0}{2} \leq \mathbb{E}\left[f_{\boldsymbol{w}}(X)^2 (y(X) - f_{\boldsymbol{w}}(X))^2\right] \geq 2L_0,$$

which ends the proof.

### C.4   Hitting time for the tied dynamics

We are now ready to prove Theorem 1 (as well as part of Theorem 2). In light of Theorem 4, it suffices to compute the hitting time $\tau_{\eta}$ of the tied dynamics (30). Since $\phi_{\text{tied}}(m) = 2C_{\text{SIE}} \times (\mathbf{1}_L, \ldots, \mathbf{1}_L) m^{\text{SIE}-1} + O(m^{\text{SIE}})$, there exists an $\eta_{\text{tied}} > 0$ such that if $m < \eta_{\text{tied}}$,

$$c_{\text{SIE}}(\sigma) C_{\text{SIE}} \times (\mathbf{1}_L, \ldots, \mathbf{1}_L) m^{\text{SIE}-1} < \phi_{\text{tied}}(m) < 3c_{\text{SIE}}(\sigma) C_{\text{SIE}} \times (\mathbf{1}_L, \ldots, \mathbf{1}_L) m^{\text{SIE}-1}$$

Combining this with the bound of Lemma 6, we find that whenever $m(t) \leq \eta$ for some constant $c > 0$,

$$\frac{dm}{dt} \geq c \cdot C_{\text{SIE}} \times (\mathbf{1}_L, \ldots, \mathbf{1}_L) m^{\text{SIE}-1} - C\gamma dm$$

For any $\delta > 0$, there exists a constant $\kappa(\delta)$ such that with probability at least $1 - \delta$,

$$m(0) \geq \frac{\kappa(\delta)}{\sqrt{d}}.$$

As a result, when $\gamma \leq \kappa(\delta)^{\text{SIE}-1} C^{-1} d^{-\frac{\text{SIE}}{2}+1}$, then for $0 \leq t \leq \tau_\eta$

$$\frac{dm}{dt} \geq c' \cdot C_{\text{SIE}} \times (\mathbf{1}_L, \ldots, \mathbf{1}_L) m^{\text{SIE}-1}$$

This implies:

- when SIE $= 1$,

$$m(t) \geq m_0 + \frac{C_{\text{SIE}} \times (\mathbf{1}_L, \ldots, \mathbf{1}_L)}{2} t, \quad \text{hence} \quad \tau_\eta \leq \frac{C'\eta}{C_{\text{SIE}} \times (\mathbf{1}_L, \ldots, \mathbf{1}_L)};$$

- when SIE $= 2$,

$$m(t) \geq m_0 \exp\left(\frac{C_{\text{SIE}} \times (\mathbf{1}_L, \ldots, \mathbf{1}_L)}{2} t\right), \quad \text{hence} \quad \tau_\eta \leq \frac{C' \ln(d)}{C_{\text{SIE}} \times (\mathbf{1}_L, \ldots, \mathbf{1}_L)};$$

- when SIE $\geq 2$,

$$m(t) \geq \left(m(0)^{2-\text{SIE}} - \frac{C_{\text{SIE}} \times (\mathbf{1}_L, \ldots, \mathbf{1}_L)}{2} t^{\text{SIE}-2}\right)^{-\frac{1}{\text{SIE}-2}}, \quad \text{hence} \quad \tau_\eta \leq \frac{C' d^{\frac{\text{SIE}}{2}-1}}{C_{\text{SIE}} \times (\mathbf{1}_L, \ldots, \mathbf{1}_L)}.$$

The bound on $\gamma$ above is always weaker (up to constant factors) than the bound $\gamma \leq C(d\tau_d)^{-1}$ of Theorem 4, and hence we always have

$$t_\eta^+ \leq Cd\tau_\eta^2,$$

which corresponds to the statement of Theorem 1.

On the other hand, for any $t \geq 0$ we have

$$\frac{dm}{dt} \leq 3C_{\text{SIE}} \times (\mathbf{1}_L, \ldots, \mathbf{1}_L) m^{\text{SIE}-1},$$

which by the same reasoning implies that the bounds on $\tau_\eta$ that we obtained are sharp up to constants.

## C.5  Hitting time for untied dynamics

We are now ready to finish the proof of Theorem 2. Since

$$\phi_{\text{untied}}(\boldsymbol{m}) = 2c_{\text{SIE}}(\sigma) C_{\text{SIE}} \times (I_L, \boldsymbol{m}, \ldots, \boldsymbol{m}) + O(\|\boldsymbol{m}\|^{\text{SIE}}),$$

for small enough $\eta > 0$ we have for $t \leq \tau_\eta$

$$\frac{d\|\boldsymbol{m}\|}{dt} \leq C \cdot C_{\text{SIE}} \times (\boldsymbol{m}, \ldots, \boldsymbol{m}) + \|C_{\text{SIE}}\| \|m\|^{\text{SIE}-1} \leq (C+1) \|C_{\text{SIE}}\| \cdot \|\boldsymbol{m}\|^{\text{SIE}-1}.$$

Recall that the hitting time $\tau_{\eta,\text{untied}}$ corresponds to $\|m\|$ hitting the threshold $\eta\sqrt{L}$. As a result, since the dependency in $\eta$ is of leading order only for SIE $= 1$, we find

$$\tau_{\eta,\text{untied}} \geq \frac{c}{\|C_{\text{SIE}}\|} \begin{cases} \eta\sqrt{L} & \text{if SIE} = 1 \\ \log(d) & \text{if SIE} = 2 \\ d^{\frac{\text{SIE}}{2}-1} & \text{if SIE} \geq 3 \end{cases}$$

Since the gain satisfies

$$\text{gain} \asymp \left(\frac{\tau_{\eta,\text{untied}}}{\tau_\eta}\right)^2,$$

this proves the first part of Theorem 2.

For the second part, assume that $C_{\text{SIE}}$ is *odeco*, hence there exists $\lambda_1, \ldots, \lambda_L$ and orthogonal vectors $\boldsymbol{v}_1, \ldots, \boldsymbol{v}_L$ such that

$$C_{\text{SIE}} = \sum_{i=1}^{L} \lambda_i \boldsymbol{v}_i^{\otimes \text{SIE}}.$$

We assume that the $\lambda_i$ are ordered by absolute value, so that $\|C_{\text{SIE}}\| = |\lambda_1|$ Letting $m^{(1)} = \langle \boldsymbol{m}, \boldsymbol{v_1} \rangle$, we have

$$\langle \boldsymbol{v}_1, C_{\text{SIE}} \times (I_L, \boldsymbol{m}, \ldots, \boldsymbol{m}) \rangle = C_{\text{SIE}} \times (\boldsymbol{v}_1, \boldsymbol{m}, \ldots, \boldsymbol{m}) = \lambda_1 (m^{(1)})^{\text{SIE}-1}$$

For small enough $\eta$, this implies that

$$\frac{dm^{(1)}}{dt} \geq c \|C_{\text{SIE}}\| (m^{(1)})^{\text{SIE}-1}$$

Since $\|m\| \geq m^{(1)}$ by the Cauchy-Schwarz inequality, the hitting time $\tau_{\eta,\text{untied}}$ is at most that of $m^{(1)}$, and hence

$$\tau_{\eta,\text{untied}} \leq \frac{C}{\|C_{\text{SIE}}\|} \begin{cases} \eta \sqrt{L} & \text{if SIE} = 1 \\ \log(d) & \text{if SIE} = 2 \\ d^{\frac{\text{SIE}}{2}-1} & \text{if SIE} \geq 3 \end{cases}$$

This closes the upper bound of Theorem 2.

### C.6 Proof of 1

Let $R^{\text{new}}(\boldsymbol{e}, m)$, $R(m)$ be the reduced population losses for the model with and without positional encoding, respectively, so that $R(m) = R^{\text{new}}(\boldsymbol{0}, m)$. Then

$$\frac{\partial^k R^{\text{new}}}{\partial m^k}(\boldsymbol{0}, 0) = \frac{d^k R}{dm^k}(0),$$

and hence $\nabla^k R(0) \neq 0$ implies that $\nabla^k R^{\text{new}} \neq 0$.

## D   Sequence Information Exponent Beyond attention

In the main paper, we discussed the learning of a generic Sequence Single Index model with a parametrized model like Equation (1), with the goal of modelling the attention mechanism learning. The theory of Sequence Information Exponent goes beyond this, and can be used to understand the sample complexity of the learning of a generic sequence single-index model both as a target and as a trained model. In this Appendix, we focus on a particular choice of positional encoding that breaks the even symmetry of the model, allowing attention to weakly recover odd SIE targets. In particular, we consider a dynamical positional encoding of the form:

$$P_i = \frac{c_i}{\sqrt{d}} \boldsymbol{w} + \tilde{P}_i \qquad \text{with } \tilde{P}_i \in \mathbb{R}^{L \times d} \text{ a fixed vector.} \tag{32}$$

$\boldsymbol{c} \in \mathbb{R}^L$ is a fixed vector of coefficients. We call this special version of positional encoding *injected positional encoding*. The trained model becomes:

$$f_{\boldsymbol{w}}(X) = R \left[ \text{softmax} \left( \left( X + \frac{\tilde{P}}{\sqrt{d}} \right) \boldsymbol{w} \boldsymbol{w}^\top \left( X + \frac{\tilde{P}}{\sqrt{d}} \right)^\top + \boldsymbol{c} \boldsymbol{c}^\top \right) \right], \tag{33}$$

and it has now a non-zero odd Hermite expansion. In Figure 6 we show examples of population losses with odd targets, learned with the model in Equation (33). The injected positional encoding breaks the symmetry, as the population risk plots show.

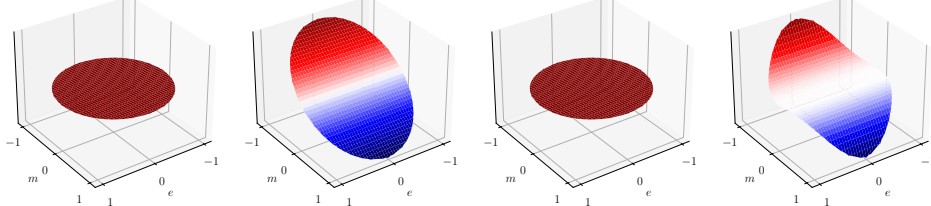

Figure 6: Population loss of the model in Equation ([33](#)) with odd targets. (left) $g(z_\star) = \mathrm{He}_1(z_{\star,1}) + \mathrm{He}_1(z_{\star,2})$, no positional encoding; (center-left) $g(z_\star) = \mathrm{He}_1(z_{\star,1}) + \mathrm{He}_1(z_{\star,2})$, injected positional encoding; (center-right) $g(z_\star) = \mathrm{He}_3(z_{\star,1}) + \mathrm{He}_3(z_{\star,2})$, no positional encoding; (right) $g(z_\star) = \mathrm{He}_3(z_{\star,1}) + \mathrm{He}_3(z_{\star,2})$, injected positional encoding. $L = 2$, $\tilde{P} = 0$ in order to isolate the effect of the injection.

## E  Further analysis on the effect of sequence length

The aim of this Appendix is to clarify and give further examples on Theorem [2](#). Here we provide an intuitive explanation of the result, while the mathematical details can be found in Appendix [C](#).

Let's start from the main result on the gain, that can be broken down in three parts:

$$\text{gain} \sim (\text{gain at costant } \gamma) \cdot \left(\frac{\gamma_{\text{tied}}}{\gamma_{\text{untied}}}\right) \cdot (\text{special factor for SIE} = 1) \tag{34}$$

**The gain constant $\gamma$**  This speedup comes from the different structure of the two networks, the tied one can built up correlation faster than the tied one because it compose $L$ different signals. The strength of this effect is strongly dependent on the target function, the overall result is

$$\text{gain at constant } \gamma \sim \frac{C_{\text{SIE}} \times (\mathbf{1}, \dots, \mathbf{1})}{\|C_{\text{SIE}}\|_{\text{op}}}. \tag{35}$$

**The ratio of the learning rates**  In Appendix [C](#), we showed that the the learning rate can grow with the sequence length $L$ at most as

- for the tied network
$$\gamma_{\text{tied}} \lesssim C_{\text{SIE}} \times (\mathbf{1}, \dots, \mathbf{1}),$$

- for the untied network
$$\gamma_{\text{untied}} \lesssim \|C_{\text{SIE}}\|_{\text{op}}.$$

It is clear that saturating the bounds above leads to a factor

$$\frac{\gamma_{\text{tied}}}{\gamma_{\text{untied}}} \lesssim \frac{\max \gamma_{\text{tied}}}{\max \gamma_{\text{untied}}} \sim \frac{C_{\text{SIE}} \times (\mathbf{1}, \dots, \mathbf{1})}{\|C_{\text{SIE}}\|_{\text{op}}}, \tag{36}$$

that is exactly the same as the gain at constant $\gamma$. Obviously the gain measure is fair only if the learning rates bounds are saturated, but our result predict the speed-up factor even in the case where the optimal learning rates are not used.

**The special factor for $\text{SIE} = 1$**  The case $\text{SIE} = 1$ is special because the dependence of the weak recovery time on the constant $\eta$ is not negligible, differently from the cases $\text{SIE} \geq 2$. This slows down further the learning of the untied network, by a factor $\sqrt{L}$, leading to a factor

$$\text{special factor for SIE} = \begin{cases} 1 & \text{SIE} \geq 2 \\ \sqrt{L} & \text{SIE} = 1 \end{cases} \tag{37}$$

### E.1  Example at not optimal learning rate

In Figure [3](#) we presented the speed up in the case of a target function with $\text{SIE} = 2$, and optimal learning rate for both the tied and untied network. Here we want to show that our result predicts the

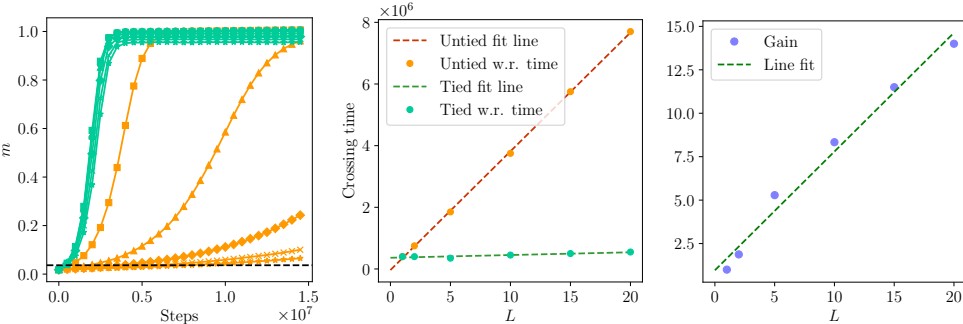

Figure 7: The gain in the case of a target function with $\mathrm{SIE} = 2$, where both cases use the same learning rate $\gamma_0 = 0.005$. (left) The evolution of the overlap, (middle) the weak recovery time and (right) the gain. The gain is proportional to $L$, as predicted.

gain well even when the learning rate are not scaled up optimally. In particular, consider the same setting as in Figure 3, but with the learning rates

$$\gamma_{\text{tied}} = \gamma_{\text{untied}} = \text{costant with } L = \gamma_0. \tag{38}$$

Using Equation (34) we can predict the gain as

$$\text{gain} \sim L \cdot 1 \cdot 1 = L. \tag{39}$$

In Figure 7 we show the result of the simulation, where we can see that the gain is indeed proportional to $L$, as predicted.

### E.2 The upper bound of learning rate scaling

In this subsection we want to show an example proving that not all scalings of the learning rate are allowed: if it grows too fast with the sequence length, the network will not be able to learn.

Let's take as example the $\mathrm{SIE} = 1$ target function

$$g(\boldsymbol{z}_\star) = \frac{1}{\sqrt{L}} \sum_{i=1}^{L} z_{\star,i}, \tag{40}$$

with the corresponding leading Hermite tensor

$$C_1 = \begin{pmatrix} 1/\sqrt{L} & \dots & 1/\sqrt{L} \end{pmatrix} \in \mathbb{R}^L. \tag{41}$$

We know that the maximum scaling for the learning rate of the untied network is

$$\gamma_{\text{untied}} \lesssim \|C_1\|_{\text{op}} = 1,$$

thus we stick with a constant learning rate $\gamma_0$ for both networks

$$\gamma_{\text{tied}} = \gamma_{\text{untied}} = \gamma_0 \sim 1. \tag{42}$$

We can use Equation (34) to have the theoretical prediction of the gain in this case:

$$C_1 \times (\boldsymbol{1}, \dots, \boldsymbol{1}) = \sqrt{L} \quad \implies \quad \text{gain} \sim \sqrt{L} \cdot 1 \cdot \sqrt{L} = L. \tag{43}$$

In Figure 8 we show the result of the simulation, where we can see that the gain is indeed proportional to $L$, as predicted. We can now ask what happens if we push the learning rate scaling of the untied network beyond the limit of Theorem 2. If we set

$$\gamma_{\text{untied}} = \gamma_{\text{tied}} = \gamma_0 \cdot L, \tag{44}$$

the ratio between the learning rates becomes now $\gamma_{\text{tied}}/\gamma_{\text{untied}} = 1/L$, and the gain

$$\text{gain} \sim \sqrt{L} \cdot \frac{1}{L} \cdot \sqrt{L} = 1. \tag{45}$$

Apparently, the untied network performance is matching the one of the tied network. Figure 9 shows the result of the simulation of such a case: the learning rate of the untied network is too high, and the network is not able for large $L$, and thus the gain diverges with a maximum value of . This plot proves our bounds on the learning rate scaling are indeed correct.

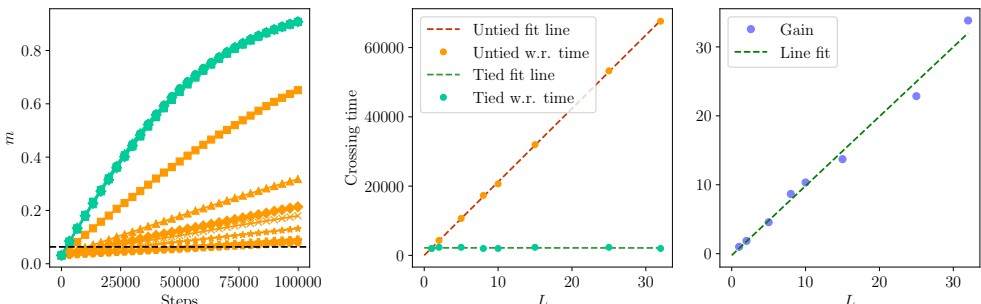

Figure 8: The gain in the case of a target function with $\text{SIE} = 1$, where both cases use the same learning rate $\gamma_0 = 0.005$. (left) The evolution of the overlap, (middle) the weak recovery time. The gain is proportional to $L$, as predicted. $d = 1000$, $\sigma = \text{ReLU}$

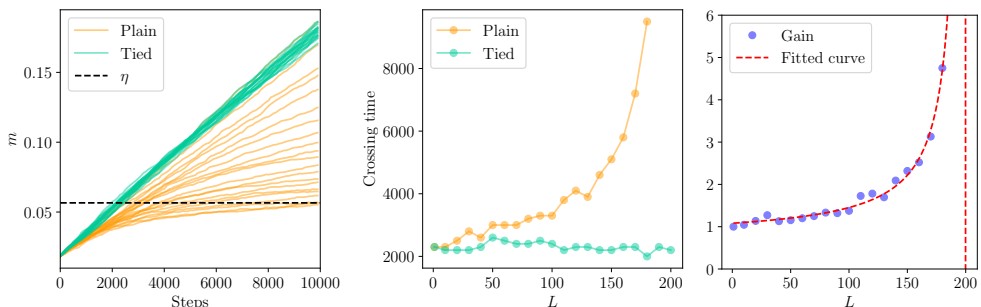

Figure 9: The gain in the case of a target function with $\text{SIE} = 1$, where both cases use the same learning rate $\gamma_0 = 0.005$. (left) The evolution of the overlap, (middle) the weak recovery time and (right) the gain. The gain diverges, as predicted. $d = 1000$, $\sigma = \text{ReLU}$

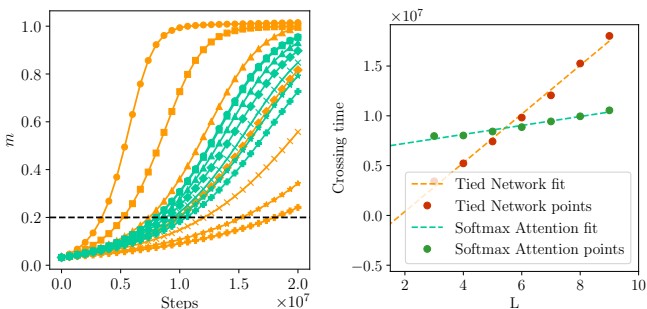

Figure 10: comparison between the learning of the tied network with squared activation (orange), namely linear attention, and the single-layer softmax attention(green) (left) The evolution of the overlap; each symbol is a different value of $L$. (right) the weak recovery time. The two networks learn at the same rate, but with different costants. $g(z_\star) = \sum_{i=1}^{L} \text{He}_2(z_{\star,i})$, $d = 1000$, $\sigma = \text{ReLU}$

### E.3 Equivalence between Single-Layer attention and Tied network

In the main paper we proved that the tied network model is a generalization of the single-layer attention model. In particular, if the activation function is $\sigma(x) = x^2$, the tied network is equivalent to the single-layer *linear* attention. We also claimed that adding a non-linearity to the attention can only improve the performance of the network, although not affecting the scaling with the sequence length. In Figure 10 we show the comparison between the learning of the tied network with squared activation (orange), namely linear attention, and the single-layer softmax attention(green). The two networks learn at the same linear rate, but with different growth constants. Softmax attention performs better than the tied network with squared activation for sufficiently large sequence lengths.

### E.4 Pathological cases

Theorem 2 shows that there could be cases where the gain from learning is 0, meaning that the tied network cannot learn anything, while the untied one possibly can. In particular the degenerate condition happens when

$$C_{\mathrm{SIE}} \times (\mathbf{1}, \dots, \mathbf{1}) = 0, \tag{46}$$

where Theorem 2 guarantees that the gain is 0. Examples of such pathological cases are:

- SIE = 1: a possible pathological target could be

$$g(\boldsymbol{z}_\star) = z_{\star,1} - z_{\star,2}.$$

  In this case the leading term in Hermite expansion is

$$C_1 = \begin{pmatrix} 1 \\ -1 \end{pmatrix} \quad \text{and consequently } C_1 \times \mathbf{1} = 0.$$

- SIE = 2: anlogously, a possible pathological target could be

$$g(\boldsymbol{z}_\star) = z_{\star,1}^2 - z_{\star,2}^2.$$

  In this case the leading term in Hermite expansion is

$$C_2 = \begin{pmatrix} 1 & 0 \\ 0 & -1 \end{pmatrix} \quad \text{and consequently } C_2 \times \begin{pmatrix} 1 & 1 \\ 1 & 1 \end{pmatrix} = 0.$$

In Figure 11 we show the pathological case for SIE = 1 and SIE = 2. The untied network is not able to learn the target function because of the symmetry: the tied network is by design symmetric, while the target is constructed to be as antisymmetric as possible in the sequence length. The untied network instead is able to learn the target function, because the weights are not constrained to be equal, thus the symmetry is broken by the initialization.

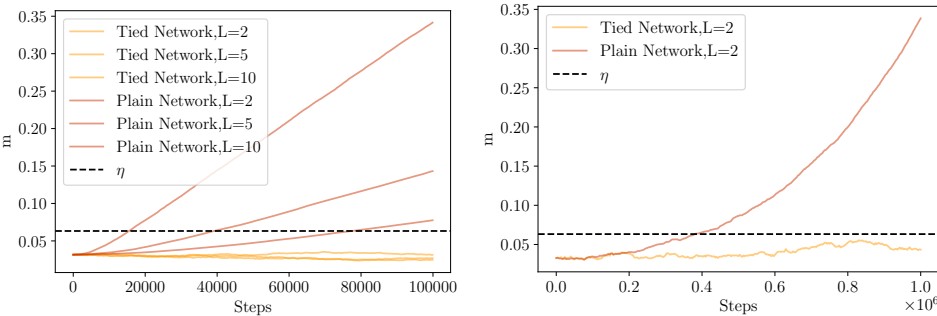

Figure 11: Pathological cases for SIE = 1 (left) and SIE = 2 (right). The untied network is able to learn the target function, while the tied one cannot.

The degeneracy of the tied network could be solved by weighting randomly the neurons. In that case, almost surely we have

$$C_{\mathrm{SIE}} \times (\mathbf{1}, \dots, \mathbf{1}) \neq 0. \tag{47}$$

Although this is a possible solution, we leave the study of the effect of random weights fro symmetry breaking for future work.

# F Further discussion on the positional-semantic transition

## F.1 SIE of model (18)

Definition 2 of the sequence information exponent does not apply directly to the model (18) because the model has a non-scalar output. However, we can still extend the concept of SIE to this case by looking at the update rule of sufficient statistics around initialization.

The population loss function is given by

$$
R(e, m) = \mathbb{E}_{\boldsymbol{z}, \boldsymbol{z}_\star \sim \mathcal{P}(e,m)} \left[ \left\| (1-\omega) \operatorname{softmax}\left( \boldsymbol{z}_\star \boldsymbol{z}_\star^\top \right) + \omega \operatorname{softmax} \left[ \begin{pmatrix} a^2 & -a^2 \\ -a^2 & a^2 \end{pmatrix} \right] - \operatorname{softmax}\left( \boldsymbol{z}\boldsymbol{z}^\top \right) \right\|_F \right]
$$
$$
= \mathbb{E}_{\boldsymbol{z}, \boldsymbol{z}_\star \sim \mathcal{P}(e,m)} \left[ \mathcal{L}(\boldsymbol{z}, \boldsymbol{z}_\star) \right]
$$

(48)

with

$$
\mathcal{P}(e, m) \equiv \mathcal{N}\left( \begin{pmatrix} 0 \\ 0 \\ e \\ -e \end{pmatrix}, \begin{pmatrix} 1 & 0 & m & 0 \\ 0 & 1 & 0 & m \\ m & 0 & 1 & 0 \\ 0 & m & 0 & 1 \end{pmatrix} \right).
$$

(49)

Using parity arguments, we can easily show that the gradient of the population loss is 0 at the initialization point $(e, m) = (0, 0)$. Let $p(\boldsymbol{z}_\star, \boldsymbol{z}; e, m)$ be the probability density function associated with the distribution $\mathcal{P}(e, m)$. Then the gradient at initialization is

$$
\nabla_{(e,m)} R(e, m) \Big|_{e=m=0} = \mathbb{E}_{\boldsymbol{z}, \boldsymbol{z}_\star \sim \mathcal{P}(e,m)} \left[ \frac{\nabla_{(e,m)} p(\boldsymbol{z}_\star, \boldsymbol{z}; e, m)}{p(\boldsymbol{z}_\star, \boldsymbol{z}; 0, 0)} \mathcal{L}(\boldsymbol{z}, \boldsymbol{z}_\star) \right]
$$

(50)

$\mathcal{L}(\boldsymbol{z}_\star, \boldsymbol{z})$ is an even function of $(\boldsymbol{z}_\star, \boldsymbol{z})$, while the gradient of the probability density function is an odd function of $(\boldsymbol{z}_\star, \boldsymbol{z})$. Therefore, the product is an odd function of $(\boldsymbol{z}_\star, \boldsymbol{z})$ and the expectation is 0.

$$
\nabla_{(e,m)} R(e, m) \Big|_{e=m=0} = (0, 0).
$$

(51)

We can use the same computation technique to compute the Hessian of the population loss at initialization. This time the parity arguments does not apply, and we can show numerically that the Hessian is non-null

$$
\frac{\partial^2 R}{\partial e^2} \Big|_{e=m=0}, \quad \frac{\partial^2 R}{\partial m^2} \Big|_{e=m=0}, \quad \frac{\partial^2 R}{\partial e \partial m} \Big|_{e=m=0} \neq 0.
$$

(52)

**The information exponent** We can also compute the *information exponent* by looking at what rate $m$ grows around initialization. We use *spherical gradient descent* beacuse we assumed the norm of the vector $\boldsymbol{w}$ to be costant (as Ben Arous also does in his paper on Information Exponent):

The update rule of $\boldsymbol{w}$ is

$$
\boldsymbol{w}^{\tau+1} = \frac{\boldsymbol{w}^\tau - \gamma \nabla_{\boldsymbol{w}} \mathcal{L}}{\| \boldsymbol{w}^\tau - \gamma \nabla_{\boldsymbol{w}} \mathcal{L} \|_2} \sqrt{d},
$$

multiplying both sides by $\boldsymbol{w}_\star / d$ we get

$$
m^{\tau+1} = \frac{m^\tau - \gamma \frac{\boldsymbol{w}_\star \cdot \nabla_{\boldsymbol{w}} \mathcal{L}}{d}}{\| \boldsymbol{w}^\tau - \gamma \nabla_{\boldsymbol{w}} \mathcal{L} \|_2} \sqrt{d}.
$$

We can assume to be in the *gradient flow regime*, where the learning rate $\gamma \ll 1$

$$
\| \boldsymbol{w} - \gamma \nabla_{\boldsymbol{w}} \mathcal{L} \|_2 = \sqrt{\| w \|^2 - 2\gamma \boldsymbol{w} \cdot \nabla_w \mathcal{L} + \gamma^2 \| \nabla_w \mathcal{L} \|^2} \approx \sqrt{d} \sqrt{1 - 2\gamma \frac{\boldsymbol{w} \cdot \nabla_w \mathcal{L}}{d}},
$$

that finally lead us to

$$
m^{\tau+1} = \left( m^\tau - \gamma \frac{\boldsymbol{w}_\star \cdot \nabla_{\boldsymbol{w}} \mathcal{L}}{d} \right) \left( 1 + \gamma \frac{\boldsymbol{w} \cdot \nabla_w \mathcal{L}}{d} \right) \approx m^\tau - \gamma \frac{\boldsymbol{w}_\star \cdot \nabla_{\boldsymbol{w}} \mathcal{L}}{d} + \gamma \frac{\boldsymbol{w} \cdot \nabla_w \mathcal{L}}{d}, \quad (53)
$$

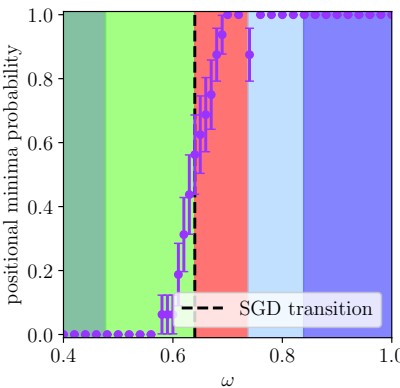

Figure 12: reproduction of Figure 5 for $d = 100$. The transition is smoother than in the $d = 1000$. Averaged over 25 runs.

where in the last step we used again the small learning rate limit.

Now we can use the chain rule for getting the gradient in terms of the derivatives we already have

$$\nabla_w \mathcal{L} = \frac{\partial \mathcal{L}}{\partial m} \cdot \nabla_{\boldsymbol{w}} m + \frac{\partial \mathcal{L}}{\partial e} \cdot \nabla_{\boldsymbol{w}} e = \frac{\partial \mathcal{L}}{\partial m} \cdot \frac{\boldsymbol{w}_\star}{d} + \frac{\partial \mathcal{L}}{\partial e} \cdot \frac{\boldsymbol{p}}{\sqrt{d}}$$

We can rearrange the equation above as

$$\frac{m^{\tau+1} - m^\tau}{\gamma/d} = -\boldsymbol{w}_\star \cdot \left( \frac{\partial \mathcal{L}}{\partial m} \cdot \frac{\boldsymbol{w}_\star}{d} + \frac{\partial \mathcal{L}}{\partial e} \cdot \frac{\boldsymbol{p}}{\sqrt{d}} \right) + \boldsymbol{w} \cdot \left( \frac{\partial \mathcal{L}}{\partial m} \cdot \frac{\boldsymbol{w}_\star}{d} + \frac{\partial \mathcal{L}}{\partial e} \cdot \frac{\boldsymbol{p}}{\sqrt{d}} \right) = -\frac{\partial \mathcal{L}}{\partial m} + m\frac{\partial \mathcal{L}}{\partial m} + e\frac{\partial \mathcal{L}}{\partial e}$$

where we used the fact $\boldsymbol{p} \cdot \boldsymbol{w}_\star \approx 0$ in high dimension (as already assumed above), and $\|\boldsymbol{w}_\star\| = d$.

We are interested in what is happening at initialization, therefore all the derivatives should be evaluated around $(e, m) = 0$. We already know that $\nabla_{e,m} \mathcal{L}\big|_{e=m=0} = 0$, so we expand the at the next order in $m$

$$\frac{m^{\tau+1} - m^\tau}{\gamma/d} = -m \left. \frac{\partial^2 \mathcal{L}}{\partial m^2} \right|_{m,e=0} + 2me \left. \frac{\partial^2 \mathcal{L}}{\partial m \partial e} \right|_{m,e=0}.$$

We can repeat the same derivation for finding the analogous equation for $e$:

$$\frac{e^{\tau+1} - e^\tau}{\gamma/d} = -e \left. \frac{\partial^2 \mathcal{L}}{\partial e^2} \right|_{m,e=0} + 2me \left. \frac{\partial^2 \mathcal{L}}{\partial m \partial e} \right|_{m,e=0}.$$

Since the hessian of he loss has always a negative eigenvalue, both these equations need $\tau = O(d \log d)$ to escape a neighborhood of initialization, leading to IE = 2.

### F.2 Numerical experiments on the transition

In this Appendix we would like to provide more details on the phase diagram of Figure 4.

In Figure 12 we reproduce Figure 5 for $d = 100$, clarifying the *finite size* effects we claimed in the main text. Smaller values of $d$ lead to a more smooth transition, since the grndient flow assumption is less valid. In the limit $d \to \infty$ we expect to see a step function.

The yellow region in the phase diagram of Figure 4 is predicted to have a unique global semantic minima, while SGD is aligned with the positional direction at initialization. The resulting trajectories are shown in Figure 13: the dynamics moves towards the minima direction, get stuck in the local flat (but not critical) region around $(e, m) = (1, 0)$, and then it moves towards the global minima. In this region the convergence is very slow. The turning point of the dynamics is a bit misplaced because of the numerical errors given by the loss integration; details on this in Appendix G.

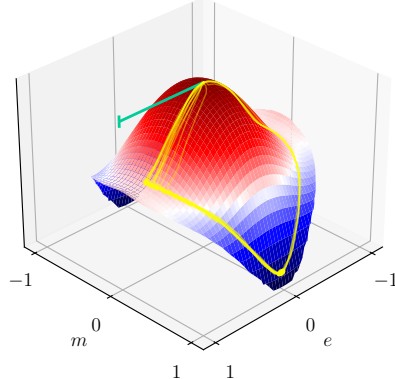

Figure 13: The loss surface in the yellow region of the phase diagram. The SGD trajectory is shown in yellow. The training moves towards the direction orthogonal to the one of global minima, ultimately slowing down the convergence.

## F.3 An alternative model without phase transition

In order to highlight the peculiarity of the model (18), we present an example of a model that does not exhibit a phase transition between the positional and semantic regimes. Let's assume that our target is a softmax attention matrix *with* positional encoding

$$y(X) = \text{softmax}\left[(X + {}^{P_\star}\!/\!\sqrt{d})\boldsymbol{w}_\star \boldsymbol{w}_\star^\top (X + {}^{P_\star}\!/\!\sqrt{d})^\top\right]$$

where

$$P_\star = \begin{pmatrix} +\boldsymbol{p}_\star \\ -\boldsymbol{p}_\star \end{pmatrix}, \qquad \boldsymbol{w}_\star = \sqrt{1 - \omega^2}\boldsymbol{w}_s + \omega \boldsymbol{p}_\star \sqrt{d} \quad \text{with} \quad \boldsymbol{w}_s \perp \boldsymbol{p}_\star.$$

$\omega$ plays the same role as in the model (18), and we can set $\omega = 0$ to get the semantic model, or $\omega = 1$ to get the positional model. The difference is that in this case the positional encoding is added to the input of the softmax function, while in the previous model it was added to the output. This change has a significant impact on the behavior of the model.

The gloabl minima of the population loss function does not transition from a positional to a semantic regime, but rather it smoothly move from the semantic to the positional regime as $\omega$ increases. We leave the details and numerical experiments for future work.

# G Numerical experiments details

All the codes used to run the experiments are available at `https://github.com/IdePHICS/Sequence-Single-Index`; where details for reproducing figures are not available in the paper, we provide the code to reproduce them in the repository.

The experiments run on a Mac Studio M2 Ultra, within at most few hours for the largest ones. The code is written in Python, using the libraries `numpy`, `scipy`, `torch` and `matplotlib`. `hydra` is used to manage the configuration files.

## G.1 Details on the integration method of squared loss

All the plot of population loss we showed are a numerical integration of the loss function. As showed in Section 1, the population loss is given by a multivariate Gaussian integral of $2L$ dimensions, where the mean and the covariance are determined by the sufficient statistics. The integral can't be computed analytically, so we use a custom numerical procedure based on the Gauss-Hermite quadrature.

Let $f \colon \mathbb{R}^{2L} \to \mathbb{R}$ be a function of $2L$ variables to be integrated, and let $\mu \in \mathbb{R}^{2L}$ and $\Sigma \in \mathbb{R}^{2L \times 2L}$ be the mean and covariance of the multivariate Gaussian distribution. The integral we want to compute

is

$$I = \int_{\mathbb{R}^{2L}} f(x) \frac{1}{(2\pi)^L} \exp\left(-\frac{1}{2}(x-\mu)^\top \Sigma^{-1}(x-\mu)\right) dx \tag{54}$$

The numerical procedure is as follows:

(i) Compute the 1D Gauss-Hermite nodes and weights for $N_{\text{int}}$ points

$$\{x_i\}_{i=1}^{N_{\text{int}}} \text{ and } \{w_i\}_{i=1}^{N_{\text{int}}}$$

where $x_i$ are the nodes given by the roots of the Hermite polynomial $H_{N_{\text{int}}}(x)$ and $w_i$ are the corresponding weights, computed as

$$w_i = \frac{2^{N_{\text{int}}}\sqrt{\pi}}{N_{\text{int}}!} \frac{1}{H'_{N_{\text{int}}}(x_i)^2}$$

(ii) Compute the $2L$ dimensional nodes and weights by taking the Kronecker product of the 1D nodes and weights

$$\{X_i\}_{i=1}^{N_{\text{int}}^{2L}} = \bigotimes_{l=1}^{2L} \{x_i\}_{i=1}^{N_{\text{int}}} \text{ and } \{W_i\}_{i=1}^{N_{\text{int}}^{2L}} = \bigotimes_{l=1}^{2L} \{w_i\}_{i=1}^{N_{\text{int}}}$$

(iii) Let $T$ be the Cholesky decomposition of $\Sigma$, i.e. $\Sigma = T^\top T$. We can then change the variable to $y = T^{-1}(x - \mu)$ and compute the integral as

$$I = \int_{\mathbb{R}^{2L}} f(x) \frac{1}{(2\pi)^L} \exp\left(-\frac{1}{2}(x-\mu)^\top \Sigma^{-1}(x-\mu)\right) dx = \int_{\mathbb{R}^{2L}} f(Ty+\mu) \frac{1}{(2\pi)^L} \exp\left(-\frac{1}{2}y^\top y\right) dy$$

(iv) The integral can then be approximated, as

$$I \approx \sum_{i=1}^{N_{\text{int}}^{2L}} \left(\prod_{l=1}^{2L} W_{i,l}\right) f(TX_i + \mu)$$

The precision of the integral is obviously regulated by the number of points $N_{\text{int}}$ we use. In our experiments, we used $N_{\text{int}} = 17$, while for the phase diagram in Figure 4 we used $N_{\text{int}} = 19$. Some effects of this integration error are visible in Figure 13.

