# OpenReview forum: "Asymptotics of SGD in Sequence-Single Index Models and Single-Layer Attention Networks"
_NeurIPS.cc/2025/Conference — NeurIPS 2025 poster_

### Official Review · Reviewer_GTHc · 2025-07-02

**Clarity:** 2
**Significance:** 1
**Originality:** 3
**Rating:** 3
**Confidence:** 4

**Summary:**

The authors study the ability of a one-layer attention mechanism to learn a sequence-single index (SSI) model, which generalizes the classic single-index model to sequence inputs. Specifically, they approximate one-pass SGD by gradient flow and analyze the sample complexity required for weak recovery. They also investigate how positional encoding and sequence length affect sample complexity.

**Questions:**

(Q1) In line 116, you state: “Note that while the model might appear as too simplified because of the lack of correlation between the tokens, we will show that it is sufficient to capture the main features of sequence models.” Which specific results in the paper demonstrate this claim?

(Q2) How exactly does the gradient flow defined in Equation (11) relate to the sample complexity?

(Q3) In Lemma 2, it is claimed that adding positional encoding lowers the information exponent. Is this a formal statement? Since the information exponent is a property of the test function $g$, I would expect it not to change even if the model is modified. (If I am misunderstanding this, please clarify.)

(Q4) In line 237, you state: “In terms of performance, taking a general activation $\sigma$ will at worst be the same of the attention mechanism originally considered, if not better.” What is the justification for this claim?

(Q5) In Section 3, why did you consider the linear attention model instead of eq.(1)?

**Ethical Concerns:**

["NO or VERY MINOR ethics concerns only"]

**Final Justification:**

This paper investigates the SGD dynamics of Transformers in the setting where the true function is given by the sequence single-index (SSI) model. The SSI model extends the single-index model to sequence inputs and includes cases where the teacher model employs a rank-one tied-weight single-layer attention matrix.

The authors are to be commended for building a rigorous theory of SGD dynamics on top of prior theoretical work on the SSI model. However, I assign a rating leaning toward rejection due to (1) limited insight into practical applicability, and (2) the reliance of the positional-encoding theory on comparisons with models of restricted expressive capacity. My initial rating was “Reject,” but after the authors clarified the connection to previous work and addressed theoretical ambiguities during the discussion period, I raised it to “Borderline Reject.”

**Limitations:**

No, the authors do not discuss limitations of their study within the paper. At the very least, I believe the issues I raised in (W3) and (W4) should be explicitly discussed.

**Paper Formatting Concerns:**

Nothing to comment here.

**Quality:**

2

**Strengths And Weaknesses:**

**Strengths:**

(S1) Extending the single-index model to sequence inputs is a unique contribution.

(S2) The paper does not merely analyze sample complexity, but also provides both theoretical and empirical discussions on how sequence length and positional encoding affect learning. The use of concrete examples to supplement the explanations is commendable.

**Weaknesses:**

(W1) Although the paper frames this work as an extension of the single-index model to sequences, the technical differences in the proofs and analysis seem relatively minor compared to standard single-index models.

(W2) It is unclear what fundamental insights the authors aim to uncover through the SSI setting. For example, when analyzing two-layer neural networks for learning single-index models, the goal is to study the feature learning capability of neural networks. In contrast, the SSI model is simply a generalization of the single-index model by introducing functions along the sequence dimension, and the motivation for studying such a model is not clearly articulated.

(W3) The model defined in Equation (1) is an even function, so if the true target function is not even, then even after achieving weak recovery, it is not guaranteed that the model can fit the true function. To achieve this, one would typically need to add a feed-forward layer after the attention layer, but doing so would change the gradients flowing through the attention layer. Thus, the model studied in this work is subject to strong simplifications for tractability.

(W4) The Sequence Information Exponent is defined via products of Hermite polynomials, but the necessity of this choice is not sufficiently explained. In standard single-index models, the Hermite polynomials serve as an orthogonal basis for Gaussian inputs, which justifies their use. However, it is not discussed whether similar justification holds in the SSI setting. Additionally, in classic single-index models, the (C)SQ lower bound characterizes learning limits in terms of the Information Exponent, but it remains unclear whether a similar limit applies to the SSI setting. Therefore, the practical significance of the upper bounds derived in this paper is not clear.

---

> ### Author Rebuttal · Authors · 2025-07-30
>
> We thank the reviewer for the detailed review.
>
> ### Weaknesses
> > Although the paper frames this work as an extension of the single-index model to sequences, the technical differences in the proofs and analysis seem relatively minor compared to standard single-index models.
>
> The primary contribution of our work is indeed conceptual: it lies in formulating and analyzing a sequence-structured analogue of the classical single-index model that bridges the gap between established theoretical frameworks and attention-based architectures used in practice.
>
> While our mathematical results build on those developed for standard single-index models, the geometry of the population loss becomes significantly more intricate, requiring new sufficient statistics that jointly capture semantic and positional alignment. This richer structure underpins phenomena such as alignment failure, sample complexity phase transitions, and the impact of positional encoding --- none of which arise in the classical setting.
>
> We view this conceptual reframing as important precisely because it enables technically grounded insights into the learning dynamics of simplified attention models, while staying mathematically tractable. Even if the technical machinery appears familiar, its application yields qualitatively new behaviors specific to sequence models.
>
> Moreover, we believe it is a particularly exciting aspect of this work that the tools developed for studying single-index models can be adapted and extended to provide rigorous insight into the training dynamics of attention-based architectures. It is a first step towards a mathematical understanding of the role of sequential structure in learning.
>
> > It is unclear what fundamental insights the authors aim to uncover through the SSI setting. .... the SSI model is simply a generalization of the single-index model by introducing functions along the sequence dimension, and the motivation for studying such a model is not clearly articulated.
>
> We respectfully disagree with this comment. While the motivation may not have been sufficiently emphasized in the manuscript—a point we will address in the revision—it is scientifically strong: the SSI model provides a minimal yet expressive framework to rigorously analyze how sequence structure and attention mechanisms affect learning under SGD.
>
> Just as classical single-index models isolate the core of feature learning in two-layer networks, the SSI model isolates key aspects of attention-based models—such as positional encoding—in a tractable setting. It enables rigorous analysis of phenomena like sample complexity scaling with sequence length, the effect of positional encoding, and alignment phase transitions, none of which are accessible in classical single-index models. A very concrete and direct consequence of our results is to provide a class of tasks for which one can rigorously establish the *computational* benefits of sequence length for attention-based architectures over their fully connected counterpart. This is among the first results in this direction in the existing literature.
>
> Far from being an arbitrary generalization, the SSI model is a deliberate abstraction aimed at uncovering what makes learning from sequences distinct. It opens a path toward a theoretical understanding of attention-based learning grounded in established tools.
>
> > The model defined in Equation (1) is an even function, so if the true target function is not even [...] the model studied in this work is subject to strong simplifications for tractability.
>
> We have dedicated appendix D to non-even targets, that can fully recover a larger set of functions.
>
> Moreover, in the single-index literature it is frequent to consider  the weak recovery phase of the learning, since is the one dominating the sample complexity; as the reviewer mentioned, surpassing the low expressivity of our model is not hard in practice, yet makes the theoretical analysis more complex.
>
> We will emphasize this aspect more and state our focus on even functions as a limitation of our work and the full treatments of generic functions as a direction of future work.
>
> > The Sequence Information Exponent is defined via products of Hermite polynomials, but the necessity of this choice is not sufficiently explained. In standard single-index models, the Hermite polynomials serve as an orthogonal basis for Gaussian inputs, which justifies their use.
>
> An equivalent justification is provided in Appendix B, and will be further clarified in the main text. As the Hermite polynomials are an orthogonal basis under the one dimensional standard Gaussian measure, all possible products of Hermite polynomials $\mathrm{He}_ {k_1}(x_1)\dots\mathrm{He}_{k_L}(x_L)$ are an orthogonal basis under the $L$-dimensional measure.
>
> > In classic single-index models, the (C)SQ lower bound characterizes learning limits in terms of the Information Exponent, but it remains unclear whether a similar limit applies to the SSI setting. Therefore, the practical significance of the upper bounds derived in this paper is not clear.
>
> We appreciate the reviewer’s point and agree that establishing (C)SQ lower bounds for the SSI setting is an important open direction. Our goal in this paper is to lay the groundwork for such results by introducing the SSI framework and deriving sharp upper bounds based on the Sequence Information Exponent (SIE), in close analogy to what was done for classical single-index models. While we do not yet provide a corresponding lower bound, we believe our analysis highlights the structural features—such as the role of the Hermite expansion and the geometry of sufficient statistics—that are likely to be relevant for future hardness results. In this sense, our contribution is a necessary step toward a complete information-theoretic understanding of learning in sequence-structured problems.
> We will dicuss this in limitations and future directions.
>
> ### Questions
> > “Note that while the model might appear as too simplified because of the lack of correlation between the tokens, we will show that it is sufficient to capture the main features of sequence models.” Which specific results in the paper demonstrate this claim?
>
>  We will reword to: "we will show that it is sufficient to capture some of the key properties of sequence models."
>
> In particular, we were able to demonstrate a transition between the positional and semantic regimes, modeling both of these properties that distinguish sequential models from unstructured ones.
> The model also allowed us to study the sequence length dependence of the sample complexity.
>
> > How exactly does the gradient flow defined in Equation (11) relate to the sample complexity?
>
> The precise relationship between the gradient flow of (11) and the sample complexity is stated in Theorem 4 (Appendix C). In short, the dynamics of $m^{\tau}$ closely follow those of $m(\gamma \tau)$, where $m(t)$ is the overlap of the solution $w(t)$ to (11), as long as $\gamma$ is small enough. Choosing the largest possible $\gamma$ such that the ODE approximation stays valid allows us to relate the hitting time of the ODE (11) to the one of the process (2). We will add a reference to Theorem 4 in the main text.
>
> > In Lemma 2, it is claimed that adding positional encoding lowers the information exponent. Is this a formal statement? Since the information exponent is a property of the test function , I would expect it not to change even if the model is modified.
>
> Thank you for pointing this out: the information exponent is indeed a property of the target function. The intention of the lemma was to express that the sample complexity of a model with positional encoding is asymptotically less than or equal to that of the same model learning the same target function without positional encoding. The model with positional encoding effectively behaves as if the target has a $\text{SIE}_\text{positional} \le \text{SIE}$, in the sense of Theorem 1. We agree that this section needs to be reformulated for clarity: we will change Lemma 1 accordingly and include the formal statement in Appendix C.
>
> > “In terms of performance, taking a general activation $\sigma$ will at worst be the same of the attention mechanism originally considered, if not better.” What is the justification for this claim?
>
> We showed that our model with linear attention is equivalent to the one in Equation (14) with $\sigma = x^2$. By allowing a generic activation function $\sigma$, our analysis encompasses a broad class of models, including the specific case $\sigma = x^2$. This generalization makes it possible to explore whether other activation functions may outperform $\sigma = x^2$ in terms of weak recovery speed, which is the metric we use to evaluate performance.
>
> > In Section 3, why did you consider the linear attention model instead of eq.(1)?
>
> Linear attention is primarily a way to draw a connection between attention mechanisms and tied/untied neural networks. Building on the previous point, the model considered in Section 3 is significantly more general than linear attention: by allowing any activation function $\sigma$, we can emulate sequential models that go beyond the standard $\mathrm{softmax}$ aggregation strategy. We will clarify this point in the final version.

---

> > ### Comment · Reviewer_GTHc · 2025-08-05
> >
> > Thank you to the authors for their detailed response.
> >
> > > While our mathematical results build on those developed for standard single-index models,
> > >
> >
> > I do not deny that extending the analysis of single-index models to sequences introduces certain technical differences (while perhaps not major). However, I expect the authors to more clearly elaborate on what specific technical differences arise and, conversely, which aspects remain consistent. Clarifying the overlapping parts not only helps delineate the contributions, but also illuminates the fundamental relationship between SSI and standard single-index models.
> >
> > In addition, I believe the authors should comment on how the sample complexity result for learning SSI (Theorem 1) corresponds to results in the standard single-index setting. Since there are multiple conceivable ways to extend standard single-index models to sequence inputs, correspondence between SSI and single-index would aid our understanding on how SSI stands among such extensions.
> >
> > Moreover, unlike the standard single-index case, the input dimensionality in SSI is not $d$ but $dL$. Therefore, in order to clearly delineate the differences, I believe the dependence on sequence length $L$ should be explicitly clarified.
> >
> > > We respectfully disagree with this comment. While the motivation may not … the SSI model provides a minimal yet expressive framework to rigorously analyze how sequence structure and attention mechanisms affect learning under SGD.
> > >
> >
> > I agree that analyzing the SSI model allows for a tractable framework, yielding insights that would be hard to obtain for more complex models. However, while the standard single-index model was proposed under a clear motivation, the SSI model appears to be an artificial extension of it, and its justification is not clear. I do not object to artificial settings yielding practically useful insights. However, it is also hard to claim that the SSI problem setting is grounded in practical concerns. For instance, the paper does not justify the assumption that a common feature vector $w_\ast$ is shared across all tokens, and this appears to be an excessive simplification. Such assumptions seem to serve the purpose of making the model analytically tractable, rather than providing insights into practically relevant scenarios. Furthermore, the experiments are only used to validate the theory under the SSI setting, and thus do not bridge theory and practice.
> >
> > One additional question: isn’t the importance of positional encoding more of an expressivity issue rather than an optimization one? The link function $g$ distinguishes all $L$ tokens, while a Transformer without positional encoding cannot distinguish between them — so it's natural that it fails to learn. While I understand that this paper focuses on optimization, even in that context, if the model cannot distinguish tokens, then the gradients also lack the information needed to tell them apart, making learning inherently difficult. Do the authors have additional clarifications on this point?
> >
> > > We have dedicated appendix D to non-even targets …
> > >
> >
> > Thank you for your response. Since Appendix D only contains empirical validation, I would encourage you to also mention the theoretical limitations in the main text.
> >
> > > An equivalent justification is provided in Appendix B, and will be further clarified in the main text. As the Hermite polynomials are …
> > >
> >
> > Thank you for the clarification. I have no further concerns on this point.
> >
> > > Our goal in this paper is to lay the groundwork for such results by introducing the SSI framework and deriving sharp upper bounds …
> > >
> >
> > I do not think providing lower bounds is the most crucial improvement needed for this work. However, since the paper positions SSI as an extension of the single-index model, lower bounds would make the correspondence between the two settings clearer. At the very least, I would expect this to be mentioned in the limitations.
> >
> > I have no further comments on the response to the questions.
> >
> > I would like to once again thank the authors for their detailed rebuttal. However, due to the remaining ambiguity around the motivation, and the unclear relationship and differences between SSI and standard single-index models, I will maintain my current rating of reject.

---

> ### Author Response · Authors · 2025-08-07
>
> > I expect the authors to more clearly elaborate on what specific technical differences arise ...
>
> On a technical level, for $k=1$ a sequence single-index model can be seen as a multi-index function $g:\mathbb{R}^{L}\to \mathbb{R}$ on the the space of tokens $\mathbb{R}^{L}$. From this connection, the expression for the sequence information exponent is closely related to the so-called leap exponent for multi-index models from (Abbe et al. 2023).
>
> The key difference is that in the context of sequence data, the arguments of the function $g$ are correlated differently than in the multi-index model.  This leads to a completely different phenomenology: for example, while for multi-index models studied in (Abbe et al. 2023) the population landscape is strictly convex (saddles + global minima), we have local minima in the SSI landscape, which captures the trade-off between semantic and positional information in the task. We will emphasise this in the revised manuscript.
>
> We reiterate that the primary interest of the SSI model is the fact that tools developed for studying multi-index models can be adapted and extended to provide rigorous insight into the training dynamics of attention-based architectures. We anticipate numerous follow-up works in that direction (e.g. those covering SQ lower bounds and other aspects that were done for multi-index models and not yet for the SSI). From this point of view, we do not think that emphasising the technical differences is particularly fruitful.
>
> > the authors should comment on how the sample complexity result for Thm 1 corresponds to results in the standard single-index setting.
>
> Can the referee be more specific about what would be the other "conceivable ways to extend standard single-index models
> to sequence inputs"? We do not see other such natural extensions and hence we do not think this constitutes a justified criticism.
>
> The correspondence here is that Theorem 1 holds for the single index model in the same form if we replace the SIE by the usual information exponent. Extension of this result to sequences, in Def. 1, is one of our main results.
>
> > Moreover, unlike the standard single-index case, the input dimensionality in SSI is not $d$ but $dL$.
>
> The dependence on the sequence length $L$ is explicit in all our results for which it is relevant, and section 3 of the paper is dedicated to this very question. We again are not sure in what this is a justified criticism.
>
> > the SSI model appears to be an artificial extension of it [...] it is also hard to claim that the SSI problem setting is grounded in practical concerns.
>
> We disagree strongly that "SSI model appears to be an artificial extension", as we formulated in the paper and our answer. Its motivation is exactly the same as for the single-index model -- to study feature learning dynamics under SGD just, but for sequence data. To put it differently, the single-index model is often motivated as a teacher generating the labels via a function that has the form or a single-layer neural network. Here it is exactly the same: a special case of SSI is a teacher that generates labels with a rank-one tied-weight single-layer attention matrix. This interpretation justifies why the feature vector $w$ is shared across tokens: this is exactly what would happen if a single-layer attention network were to learn on such data, the weights in the attention layer do not depend on the token index.
>
> Finally, we find that the SSI is no less relevant to practical scenarios than the single-index model itself.
>
> > the experiments are only used to validate the theory under the SSI setting, ...
>
> In the same way as done in a great majority of the large number of papers published on the single-index model.
>
> > isn’t the importance of positional encoding more of an expressivity issue rather than an optimization one? ....
>
> We agree with the reviewer that the presence of a positional encoding allows the transformer model to fit a richer class of functions, and therefore can be seen as a question of expressivity of the underlying hypothesis class. But just as the choice of architecture fundamentally determines the geometry of the risk landscape, so does the presence or absence of the positional encoding. And since one-pass SGD can be seen as a random approximation of gradient flow on the population risk, it is crucial to understand how the population risk is impacted by the presence of a positional encoding. This is what our results, and more specifically Lemma 1, are about.
>
> > lower bounds would make the correspondence between the two settings clearer.
>
> We agree on this, and as said in the answer, (C)SQ lower bounds are an important future direction. We want to note that there is no paper on the single-index model that would include all those results at once. Rather, there is a series of multiple papers by different groups, and we realistically expect the same here. As affirmed previously, this will be stated in the limitations.

---

> ### Comment · Reviewer_GTHc · 2025-08-08
>
> I appreciate the authors' responses.
>
> > Can the referee be more specific about what would be the other "conceivable ways to extend standard single-index models to sequence inputs"?
>
> I agree my original comment was vague. I had in mind settings where (i) each token is associated with a different feature vector, or (ii) relevant features span multiple tokens—cases more complex than a per-token single-index model. I do not think the SSI is the only possible extension. That said, I acknowledge that such alternatives would likely be analytically harder; viewed within the scope of analytical tractability, SSI can be a natural choice. My remark about alternative conceivable extensions is therefore nonessential; please feel free to disregard it. (Nonetheless, I think that it is problematic that it has limited relevance with practical scenarios, as I discuss below.)
>
> > The dependence on the sequence length $L$ is explicit in all our results for which it is relevant, and section 3 of the paper is dedicated to this very question.
>
> What I am interested in is what happens if the dependence on $L$ is made explicit in Theorem 1. I agree that the effect of $L$ is discussed in Section 3.
>
> > We disagree strongly that "SSI model appears to be an artificial extension", as we formulated in the paper and our answer. ...
>
> I understand that SSI includes the setting of rank-one tied-weight single-layer attention. However, this is only a interpretation of SSI, and it remains unclear how this connects to realistic settings. In particular, the assumption that all tokens share a common feature vector seems difficult to justify in practice. As I stated above, I agree that SSI offers a tractable extension of the single-index model. However, it appears to have limited practical relevance.
>
> > In the same way as done in a great majority of the large number of papers published on the single-index model.
>
> I agree on this point. Still, as noted above, SSI appears to have less practical significance than the standard single-index model. My suggestion regarding empirical results was aimed at strengthening the connection between SSI and realistic scenarios.
>
> > But just as the choice of architecture fundamentally determines the geometry of the risk landscape, so does the presence or absence of the positional encoding.
>
> I agree that the architecture affects the risk landscape. However, such discussions are often made under the condition of sufficient expressive capacity. Arguing that optimization is difficult when expressive capacity is lacking—that is, when the model does not even have access to the relevant information—does not seem to be a particularly strong claim.

---

> ### Author Response · Authors · 2025-08-08
>
> > I agree my original comment was vague. I had in mind settings where (i) each token is associated with a different feature vector, or (ii) relevant features span multiple tokens. I do not think the SSI is the only possible extension. That said, I acknowledge that such alternatives would likely be analytically harder; SSI can be a natural choice. My remark about alternative conceivable extensions is therefore nonessential; please feel free to disregard it.
>
> Thank you for the clarification. We now understand the question. While our paper deals with the sequence **single** index model, previous papers that considered this model [Troiani et al., 2025, Cui, 2025] actually introduced a more general multi-index version (and studied the Bayes-optimal estimator and the ERM). The variants you are proposing would require the consideration of multiple indices, analogous to the multi-index model. We believe the analysis of the SGD can also be done for those cases, but would require additional work. We will make a note about this in the revised version, this is indeed useful to clarify.
>
> > I am interested in is what happens if the dependence on $L$ is made explicit in Theorem 1.
>
> Thanks again for clarifying your question. Parsing the proof in Appendix C (around line 600), the dependence in $L$ of Theorem 1 can be written as
>
> $C_L = [C_{\text{SIE}} \times (\mathbf 1_L, \dots, \mathbf{1}_L)]^{-2}$,
>
> where $C_{\text{SIE}}$ is the first non-zero Hermite coefficient of $g$. Assuming that $g$ is normalized so that $\|C_{\text{SIE}}\| = O(1)$, for a "generic" choice of $g$ the constant $C_L$ is of order $\Theta(L^{-\text{SIE}})$.
>
> However, there can be pathological choices of $g$ (e.g. divergence-free functions) such that $C_L = 0$, and thus the tied network will not learn $g$. Examples of such functions are presented in Appendix E.4. This relates to your earlier comment on expressivity, in the sense that $g$ distinguishes the token positions in a way that the tied network is incapable to learn.
>
> We will add a pointer to this in the revised version.
>
> > The assumption that all tokens share a common feature vector seems difficult to justify in practice. As I stated above, I agree that SSI offers a tractable extension of the single-index model.
>
> We note that our paper is not the one that introduces the SSI model; this was done in references [Cui et al., 2024, Troiani et al., 2025, Cui, 2025]. Our work is the first one studying the SGD dynamics in this model (as opposed to the ERM or Bayes-optimal estimators in the previous works). This again goes along the same line of what happened for the single-index model where the Bayes-optimal estimator and ERM were studied before the SDG dynamics or before the SQ/CSQ lower bounds were established.
>
> While in our paper we focus on the setting where "all tokens share a common feature vector" the sequence-multi-index model from [Troiani et al., 2025] does not need that assumption, we thus anticipate that the SGD analysis could be done in that case as well.
>
> We also note that "all tokens share a common feature vector" is a rather generic property of transformer architectures. There are, of course many feature vectors in the SOTA transformer, but they are shared between all tokens. Indeed, none of the learnable weights in a typical transformer architecture has a dimensionality depending on the sequence length; all the tokens share all the latent variables. While we agree with the referee that this is not evident to justify, the empirical performance of such architectures should be seen as an empirical justification that such an assumption leads to useful and practically relevant models.
>
> > I agree on this point. Still, SSI appears to have less practical significance than the standard single-index model. My suggestion regarding empirical results was aimed at strengthening the connection between SSI and realistic scenarios.
>
> When it comes to real-task relevance of the model, we want to point to the histogram task experiments reported in the appendix of the reference [Cui et al., 2024] where the positional-semantic phase transition we describe appears as two possible ways of learning to solve the histogram task to which gradient descent can converge.
>
> > I agree that the architecture affects the risk landscape. However, such discussions are often made under the condition of sufficient expressive capacity. Arguing that optimization is difficult when expressive capacity is lacking does not seem to be a particularly strong claim.
>
> We also agree that it is not surprising that learning is difficult in the unrealizable setting. In our work, we do not present any negative results on the positional encoding, i.e., we do not prove that optimization cannot be achieved when $P=0$. Theorem 1 is complemented by Lemma 1 for this very reason. Could the reviewer point to a result in the paper where we "argue that optimization is difficult when expressive capacity is lacking", and $P=0$?

---

> > ### Comment · Reviewer_GTHc · 2025-08-09
> >
> > Thank you for your reply.
> >
> > > Could the reviewer point to a result in the paper where we "argue that optimization is difficult when expressive capacity is lacking", and $P=0$?
> >
> > Perhaps “argue that optimization is difficult when expressive capacity is lacking” was too strong an expression. I merely intended to refer to SIE.
> >
> > The clarification regarding the connection to the authors’ previous work has partially addressed my concerns. Based on this, I will raise my rating by one.
> >
> > Once again, I appreciate the authors’ proactive engagement with my comments during the discussion period.

---

### Official Review · Reviewer_iycP · 2025-07-02

**Clarity:** 3
**Significance:** 3
**Originality:** 4
**Rating:** 5
**Confidence:** 2

**Summary:**

This paper investigates the dynamics of Stochastic Gradient Descent (SGD) in sequence models referred to as Sequence Single-Index (SSI) models. The authors introduce the Sequence Information Exponent (SIE) as a generalization of the information exponent from single-index models, and demonstrate its close relationship with the sample complexity of SGD. The paper also explores the theoretical impact of positional encoding on the speed of SGD training. It analyzes how models equipped with attention mechanisms significantly outperform fully connected networks, which are not designed to handle sequential data, in terms of learning efficiency for sequential inputs. Furthermore, the authors show that when data contains positional and semantic information, the dynamics of SGD may not always disentangle these components, leading to a rich phase diagram that characterizes the structure of the population loss and the performance of SGD.

**Questions:**

Equation 1: Readers unfamiliar with Transformers may not understand what the variables $X$ and $P$ represent. In particular, the introduction of $P$ feels abrupt.

Line 202: What form of positional encoding is assumed here? While it is clear that it should be non-zero, it is unclear whether the authors are assuming the $\sin$ and $\cos$ form used in the original Transformer, or if any arbitrary form is acceptable.

Section 4: Is the discussion limited to the case where only $P_1$ and $P_2$ exist, that is, the case of $L=2$?

**Ethical Concerns:**

["NO or VERY MINOR ethics concerns only"]

**Final Justification:**

I see the value of the theoretical work even if it is deviated from the practical use of Transformers. The author response answers questions raised in the initial review.

**Limitations:**

There is some explanation about how this formulation is far from the practical use of Transformer. However, there is no dedicated section or paragraph explaining the limitations.

**Quality:**

4

**Strengths And Weaknesses:**

Strength

+ The paper elucidates the dynamics of SGD in SMI models.
+ It clarifies how the sequence length and positional encoding affect the convergence speed and trajectory of SGD.

Weakness

+ As a theoretical study, the approach inevitably diverges from the practical Transformer architecture. It remains unclear how applicable the insights gained in this paper are to real-world Transformer-based research.

---

> ### Author Rebuttal · Authors · 2025-07-30
>
> We thank the reviewer for the insightful comments. Below, we address the primary concerns that have been raised.
>
> ### Weakness
> > As a theoretical study, the approach inevitably diverges from the practical Transformer architecture. It remains unclear how applicable the insights gained in this paper are to real-world Transformer-based research.
>
> Our theoretical model is not the full transformer architecture, but a simplification of the attention architecture. This is almost always the case in theoretical works, where the goal is not to replicate a real setting (which is often mathematically not tractable) but rather to identify and isolate the core mechanisms —such as token interactions, positional encoding, and sequence -induced learning dynamics — in a mathematically tractable setting. Despite the simplifications, our analysis reveals rich behaviors that are qualitatively aligned with challenges observed in real-world sequence learning. We hope these insights will serve as a foundation for future theoretical work that bridges further toward practical architectures.
>
>
>
> ### Questions
> > Equation 1: Readers unfamiliar with Transformers may not understand what the variables and represent. In particular, the introduction of feels abrupt.
>
> Thank you for the feedback. We will use part of the extra space of the final version for making this step clearer; we also have Appendix A for a detailed explanation of the attention mechanism.
>
> > What form of positional encoding is assumed here? While it is clear that it should be non-zero, it is unclear whether the authors are assuming the $\sin$ and $\cos$ form used in the original Transformer, or if any arbitrary form is acceptable.
>
> From a theoretical point of view, any set of linearly independent vector could be used as positional encoding.
> When simulating or running it practically, farther apart are the positional embedding vectors, easier for the model distinguishing the positions, thus more effective it is. For most of our simulations, we use $P_1=-P_2$, in order to have the bigger separation as possible.
> In practice, any usual strategy to have a well separated positional encoding (as the one you suggest), would work.
>
> > Section 4: Is the discussion limited to the case where only $P_1$ and $P_2$ exist, that is, the case of $L=2$?
>
> No, the phase transition could be certainly observed for settings $L>2$. We stick with $L=2$ just for having a nicer visualization of what is happening, since only in this case we could reduce the problem to just 2 sufficient statistics and consequently have nice plots like Figure 7. Nevertheless the assumption $L=2$, we are able to make our point: SGD is not always converging to global minima for Sequence Single Index models.
>
> ### Limitation
> > There is some explanation about how this formulation is far from the practical use of Transformer. However, there is no dedicated section or paragraph explaining the limitations.
>
> Thanks for the precious comment. While we have described all the assumptions made for leading to our result through the text, we will make sure to add a recap Paragraph explicitly recapping them.

---

> > ### Comment · Reviewer_iycP · 2025-08-08
> > **Re: Rebuttal by Authors**
> >
> > > Our theoretical model is not the full transformer architecture, but a simplification of the attention architecture. This is almost always the case in theoretical works
> > I'm aware of the importance of theoretical work, even if it deviates from actual use. It could be a weakness of a study, so it would be nice to explain the limitations of this work to clarify the scope of this work. I don't think it can be a reason for rejection in NeurIPS conferences.
> >
> > Thank you for answering my questions. Please consider making these clearer if you think them reasonable.

---

### Official Review · Reviewer_YJDv · 2025-07-03

**Clarity:** 3
**Significance:** 3
**Originality:** 2
**Rating:** 5
**Confidence:** 3

**Summary:**

This paper analyzes the behavior of (spherical) SGD for a class of sequence models where the transformation of the input data is determined by a single vector and a link function. The trainable model is represented as a simplified attention mechanism, where the key and query matrices are tied, the attention head dimension is 1, and with an identity value matrix. The paper introduces two *sufficient statistics* (**e**, m) from which the population risk can be determined. It also generalizes the "information exponent" to sequential data, helping establish the sample complexity of SGD for the studied models. The paper further analyzes the impact of positional encodings, which can only reduce (or keep constant) the sequence information exponent, as well as the impact of the sequence length.

**Questions:**

1. You mention that "the single-layer tied attention model in eq. (1) is a particular instance of a class of sequence multi-index (SMI) models", which justifies the choice of training data (drawn from a sequence single-index model (SSI)). Could the data also be generated from a multi-index model? If so, what would that imply?

2. Can you provide an intuitive interpretation about the definition of the Sequence Information Exponent?

3. Can you clarify the statement "while the model might appear as too simplified because of the lack of correlation between the tokens, we will show that it is sufficient to capture the main features of sequence models."?

*Minor comments*

L45: Missing closing parenthesis

The paper mostly uses "sequence single-index model", but the hyphen is placed differently in the title ("sequence-single index model").

**Ethical Concerns:**

["NO or VERY MINOR ethics concerns only"]

**Final Justification:**

This submission provides a rigorous analysis of SGD on a simplified class of sequence models. While these models don't capture all the intricacies of practical Transformers, there is still non-trivial and interesting behavior to inspect. The authors addressed my limited concerns, in particular by adding nuance to their claim regarding the impact of independent inputs (or outputs).

**Limitations:**

Yes

**Quality:**

3

**Strengths And Weaknesses:**

**Strengths**

- The paper presents a detailed analysis of the behavior of (spherical) SGD for a class of sequence models related to self-attention, evaluating the SGD sample complexity and the impact of positional encodings and sequence length.

- The paper is rigorous, clearly presenting the hypotheses under which the results hold, proving key claims (although I didn't examine the proofs in the appendix in detail), presenting numerical simulations and sharing code for easily reproducible results.

**Weaknesses**

- The paper assumes independence of the inputs (and outputs) across a sequence, which is not realistic for many sequence modeling tasks. The authors mention that "while the model might appear as too simplified because of the lack of correlation between the tokens, we will show that it is sufficient to capture the main features of sequence models", but it is not clear to me how having correlated inputs/outputs would impact the claims and results.

- The paper ends abruptly, with no clear conclusion. I would like the authors to discuss the potential impact of their findings on future research directions.

---

> ### Author Rebuttal · Authors · 2025-07-30
>
> ### Weaknesses
> > The paper assumes independence of the inputs (and outputs) across a sequence, which is not realistic for many sequence modeling tasks. The authors mention that "while the model might appear as too simplified because of the lack of correlation between the tokens, we will show that it is sufficient to capture the main features of sequence models", but it is not clear to me how having correlated inputs/outputs would impact the claims and results.
>
> We agree that the assumption of independent inputs and outputs is a simplification, and we will clarify this more explicitly as a limitation in the revision. However, we would like to stress that the SSI model still captures length-wise correlations through the link function $g$, i.e. the dependence of the task outputs in the sequence inputs. Even in this simplified setting, we observe rich and nontrivial behaviors—such as alignment phase transitions, positional vs. semantic learning dynamics, and SIE-dependent sample complexity—that highlight core challenges in sequence learning. Studying correlated inputs is an important next step, and we believe our framework provides a solid foundation for doing so (see as example Cui et al. _A phase transition between positional and semantic learning in a solvable model of dot-product attention_).
>
> > The paper ends abruptly, with no clear conclusion. I would like the authors to discuss the potential impact of their findings on future research directions.
>
> Thank you for the comment. We will write a proper conclusion for the final version of the paper.
>
> ### Questions
> > You mention that "the single-layer tied attention model in eq. (1) is a particular instance of a class of sequence multi-index (SMI) models", which justifies the choice of training data (drawn from a sequence single-index model (SSI)). Could the data also be generated from a multi-index model? If so, what would that imply?
>
> Yes, in principle samples can be generated by a SMI, but as long as the attention model is rank 1, only a single direction can be recovered by the model. Our model would learn the most relevant direction (in term of lowest Hermite order), and a similar result to Theorem 1 would apply to this special direction.
> Allowing the the rank of $Q, K$ to be $>1$, all the complex effects characteristic of multi-index model would emerge. In particular, we expect to observe hierarchical learning phenomena such as _staircases_, and it might be possible to extend _leap exponents_ (Abbe et al. _SGD learning on neural networks: leap complexity and saddle-to-saddle dynamics_) to sequence model; this would be a natural future direction after this work; we will discuss this in a Future Direction section in the final revision of our paper.
>
> > Can you provide an intuitive interpretation about the definition of the Sequence Information Exponent?
>
> Every token is a gaussian vector of dimension $d$. When we project a token along a fixed direction, the resulting coefficient $z_\star$ is also gaussian variable. When $d\to+\infty$, close to initialization, all the $L$ _projected tokens_ are independent and distributed as $L$ multivariate gaussian. At this point, it is natural to decompose the target function $f^\text{SSI}$ over an orthonormal base with respect to the Gaussian distribution: that's how the Hermite polynomials appear in our derivation. Given that the target takes as inputs the $L$ projections of the tokens, the orthonormal basis elements are the product of $L$ scalar Hermite polynomials. The information exponent is nothing more than the lowest degree of the multivariate polynomials of this expansion. You can find more details in Appendix B.
>
> > Can you clarify the statement "while the model might appear as too simplified because of the lack of correlation between the tokens, we will show that it is sufficient to capture the main features of sequence models."?
>
> Answered in "Weaknesses".

---

> > ### Comment · Reviewer_YJDv · 2025-08-08
> >
> > Thank you for the detailed response.
> >
> > > We agree that the assumption of independent inputs and outputs is a simplification, and we will clarify this more explicitly as a limitation in the revision. However, we would like to stress that the SSI model still captures [...]
> >
> > Yes, there is non-trivial and interesting behavior to analyze, although the original claims might have been too strong. Thank you for clarifying this.
> >
> > > We will write a proper conclusion for the final version of the paper.
> >
> > In particular, what would be the next steps (either from you, or that you think may generally be interesting for other researchers in the field)?
> >
> > > Every token is a gaussian vector [...]
> >
> > If there is sufficient room, explaining this in the paper may be worthwhile.

---

> > > ### Author Response · Authors · 2025-08-08
> > >
> > > > In particular, what would be the next steps (either from you, or that you think may generally be interesting for other researchers in the field)?
> > >
> > > One possible future direction is to establish SQ lower bounds for the SSI setting, as has been done for the single-index setting. Another natural step would be to extend these results to sequence multi-index models by allowing a higher rank for the query/key matrices or by using different feature vectors for each token. The goal would be to study the SGD dynamics in the considered setting [Troiani et al., 2025; Cui, 2025].
> > > The SSI dynamic already exhibits richer phenomenology than plain multi-index models. For example, we demonstrated a non-convex landscape. Thus, we believe that many other research directions could emerge as the SSI framework is further developed for including multiple index vectors.

---

> > > > ### Comment · Reviewer_YJDv · 2025-08-09
> > > >
> > > > Thank you for the clarification.

---

### Official Review · Reviewer_3AjM · 2025-07-03

**Clarity:** 3
**Significance:** 3
**Originality:** 3
**Rating:** 4
**Confidence:** 2

**Summary:**

The paper investigates the training dynamics of SGD for a class of sequence single-index (SSI) models, which generalizes the classical single-index model to encompass a simplified one-layer attention model. The authors introduce a metric called the Sequence Information Exponent (SIE) to characterize the sample complexity of SGD and show that positional encoding can help SGD learn faster for certain problems. Then the authors show that a tied-weight model,  representing the attention mechanism, can learn faster than its untied counterpart with a gain proportional to $L$. Finally, the paper explores the interplay between learning semantic versus positional information, revealing a rich phase diagram that describes the different convergence behaviors of SGD.

**Questions:**

- The proof of Lemma 1 appears to be missing in Appendix C.
- Lemma 1 indicates that positional encoding does not worsen the sample complexity of SGD. Could the authors provide a more general characterization of the conditions under which positional encoding strictly improves sample complexity?

**Ethical Concerns:**

["NO or VERY MINOR ethics concerns only"]

**Final Justification:**

I'll keep my current positive score.

**Limitations:**

While the paper discusses the limitations of the assumptions made for its theoretical analysis, it would benefit from a more explicit discussion of the limitations of the architecture being studied.

**Quality:**

3

**Strengths And Weaknesses:**

Strengths
- The paper is well-written and clear
- The paper provides a strong theoretical analysis complemented by numerical experiments.

Weaknesses
- The paper considers a tied-weight and rank-$1$ attention model, limiting its applicability to real-world transformers.

---

> ### Author Rebuttal · Authors · 2025-07-30
>
> ### Weaknesses
> > The paper considers a tied-weight and rank- attention model, limiting its applicability to real-world transformers.
>
> We agree. This is analogous to the classical single-index model, which also does not capture the full complexity of multi-layer perceptrons  but has nonetheless proven foundational in understanding separations between lazy and feature learning regimes for fully-connected neural networks. Similarly, our model enables rigorous analysis while retaining key mechanisms—such as token interactions and positional encoding—and we hope it will serve as a first step towards theoretically understanding more complex attention architectures.
>
> ### Questions
> > The proof of Lemma 1 appears to be missing in Appendix C.
>
> Thank you for pointing this inattention out. The informal idea behind this Lemma is that adding the positional encoding can only increase the number of non-zero terms in the Hermite expansion, giving the chance to a lower term to correlate with the target. Thus, lowering the effective information exponent. The formal statement and proof will be added in the Appendix C of final version.
>
> > Lemma 1 indicates that positional encoding does not worsen the sample complexity of SGD. Could the authors provide a more general characterization of the conditions under which positional encoding strictly improves sample complexity?
>
> We thank the reviewer for the insightful question. A characterization of $\text{SIE}_\text{positional}$ would indeed be a natural direction for future exploration. Roughly speaking, a model having $\text{SIE} \geq k$ implies that the derivatives up to order $k$ of the loss (as a function of the overlaps) are zero _in the direction of $m$_. If any of the derivatives below order $k$ in the direction of the positional overlap $e$ are nonzero, it would imply that adding a trainable positional encoding improves the required sample complexity.
>
> ### Limitations
> > While the paper discusses the limitations of the assumptions made for its theoretical analysis, it would benefit from a more explicit discussion of the limitations of the architecture being studied.
>
> Thanks for the precious comment. While we have described all the assumptions made for leading to our result through the text, we will make sure to add a paragraph explicitly recapping them.

---

> ### Comment · Reviewer_3AjM · 2025-08-05
>
> Thanks for the response. Most of my concerns have been resolved and I'll keep my current positive score.

---

### Decision · Program_Chairs · 2025-09-17

**Decision:**

Accept (poster)

**Comment:**

The reviewers agreed that this paper addresses an important topic, is clearly written, and has strong results. Some reviewers had concerns about the technical assumptions made in the paper, and how it may affect the proposed model's applicability. I believe that the authors were able to provide detailed answers to almost all of the concerns. After going through the paper, I felt that I will have to agree with the majority of the reviews as the positives (such as originality, clarity) clearly outweigh the perceived negatives in the submission. Hence, I recommend accepting this paper.